# Environmental Conditions for Alternative Tree Cover States in High Latitudes

Beniamino Abis[1,2] and Victor Brovkin[2]

[1]International Max Planck Research School on Earth System Modelling, Hamburg, Germany
[2]Max Planck Institute for Meteorology, Hamburg, Germany

*Correspondence to:* Beniamino Abis (beniamino.abis@mpimet.mpg.de)

**Abstract.** Previous analysis of the vegetation cover from remote sensing revealed the existence of three alternative modes in the frequency distribution of boreal tree cover: a sparsely vegetated treeless state, an open woodland state, and a forest state. Identifying which are the regions subject to multimodality, and assessing which are the main factors underlying their existence, is important to project future change of natural vegetation cover and its effect on climate.

We study the link between the tree cover fraction distribution and eight globally-observed environmental factors: mean annual rainfall, mean minimum temperature, growing degree days above $0°C$, permafrost distribution, mean spring soil moisture, wildfire occurrence frequency, soil texture, and mean thawing depth. Through the use of generalised additive models, conditional histograms, and phase-space analysis, we find that environmental conditions exert a strong control over the tree cover distribution, uniquely determining its state among the three dominant modes in $\sim$95% of the cases. Additionally, we find that the link between individual environmental variables and tree cover is different within the four boreal regions here considered, namely Eastern North Eurasia, Western North Eurasia, Eastern North America, and Western North America. Furthermore, using a classification based on rainfall, minimum temperatures, permafrost distribution, soil moisture, wildfire frequency, and soil texture, we show the location of areas with potentially alternative tree cover states under the same environmental conditions in the boreal region. These areas, although encompassing a minor fraction of the boreal area ($\sim$5%), correspond to possible transition zones with a reduced resilience to disturbances. Hence, they are of interest for a more detailed analysis of land-atmosphere interactions.

## 1 Introduction

Forest ecosystems are a fundamental component of the Earth, as they contribute to its biophysical and biogeochemical processes (Brovkin et al., 2009), and harbour a large proportion of global biodiversity (Crowther et al., 2015). However, changes in species composition, structure, and function are happening in several forests around the world (Phillips et al., 2009; Lindner et al., 2010; Poulter et al., 2013; Reyer et al., 2015b). These changes originate from a combination of environmental changes, such as $CO_2$ concentration, drought, and nitrogen deposition (Hyvönen et al., 2007; Michaelian et al., 2011; Brouwers et al., 2013; Brando et al., 2014; Reyer et al., 2015a), and local drivers, both anthropogenic and not, such as forest management, wildfires, and grazing (Volney and Fleming, 2000; Malhi et al., 2008; Barona et al., 2010; DeFries et al., 2010; Bond and Midgley, 2012;

Bryan et al., 2013). Environmental and climate changes, as well as extreme events, are likely to play a more prominent role in future decades (Johnstone et al., 2010; Orlowsky and Seneviratne, 2012; Coumou and Rahmstorf, 2012; IPCC, 2013), affecting the resilience of forests - i.e., the ability to absorb disturbances maintaining similar structure and functioning (Scheffer, 2009) - and possibly pushing them towards tipping points and alternative tree cover states (IPCC, 2013; Reyer et al., 2015a), potentially inducing ecosystem shifts (Scheffer, 2009).

Increasing attention has been given to the response of ecosystems to past climate changes (Huntley, 1997; Huntley et al., 2013), and to ecosystems exhibiting potential alternative tree cover states under the same environmental conditions, as key factors to a deeper understanding of forest resilience (Scheffer, 2009; Hirota et al., 2011; IPCC, 2013; Reyer et al., 2015a). To such avail, in this paper, we investigate the relationship between environment and remotely-sensed tree cover distribution within the boreal ecozone. Through the use of generalised additive models (GAMs), conditional histograms, and phase-space analysis, we assess whether alternative stable tree cover states are possible in the boreal forest, and under which environmental conditions, as understanding the mechanisms underpinning them is a key point to assess future ecosystem changes (Reyer et al., 2015a).

The boreal forest is an ecosystem of key importance in the Earth system, as it encompasses almost 30% of the global forest area and comprises about $0.74$ trillion densely distributed trees (Crowther et al., 2015). Despite a low diversity of tree species, boreal forest's structure and composition depend on interactions between several factors, including precipitation, air temperature, available solar radiation, nutrient availability, soil moisture, soil temperature, presence of permafrost, depth of forest floor organic layer, forest fires, and insect outbreaks (Kenneth Hare and Ritchie, 1972; Heinselman, 1981; Bonan, 1989; Shugart et al., 1992; Soja et al., 2007; Gauthier et al., 2015). The boreal ecozone is highly sensitive to changes in climate and can affect the global climate system through its numerous feedbacks, the most important ones related to albedo changes, soil moisture recycling, and the carbon cycle (Bonan, 2008; Gauthier et al., 2015; Steffen et al., 2015). In fact, vegetation at high latitudes can influence albedo through its distribution and through its snow-masking effect, leading to warmer temperatures (Bonan et al., 1992). During winter, a snow-covered forest has a lower albedo than snow covered low vegetation, as tall trees mask the snow on the ground (Otterman et al., 1984; Bonan, 2008). Additionally, differences between species distributions can affect albedo in summer, as dark conifers have a lower albedo than deciduous trees or shrubs (Eugster et al., 2000). On the other hand, during the growing season, trees induce a cooling effect due to enhanced evapotranspiration with respect to low vegetation (Brovkin et al., 2009). Finally, the boreal forest acts as a carbon sink (Gauthier et al., 2015) and is responsible for an estimated $\sim$20% of the world's forest total sequestered carbon (Pan et al., 2011; Gauthier et al., 2015). The balance between these effects determines how the boreal forest influences climate, which, in turn, affects vegetation.

Despite its multiple roles in regulating climate, the dynamics of the boreal ecosystem regarding gradual changes and critical transitions are not yet understood (Bel et al., 2012; Scheffer et al., 2012). In this context, multimodality of the tree cover distribution has recently been detected within the boreal biome (Scheffer et al., 2012). An analysis of the vegetation cover from remote sensing revealed the existence of three alternative modes in the frequency distribution of boreal trees (Scheffer et al., 2012; Xu et al., 2015): a sparsely vegetated treeless state, an open woodland "savanna"-like state, and a forest state. Particularly, it has been observed that, over a broad temperature range, these three vegetation modes coexist (Scheffer et al., 2012; Xu et al.,

2015); on the other hand, areas with intermediate tree cover between these distinct modes are relatively rare, suggesting that they may represent unstable temporary states (Scheffer et al., 2012). Furthermore, it has been shown that multimodality of the tree cover does not ensue from multimodality of environmental conditions, suggesting that these three modes could represent alternative stable states acting as attractors (Scheffer et al., 2012). A stable state being the state an ecosystem will return to after any small perturbation (May, 1977). Hence, identifying which are the regions subject to multimodality, and assessing which are the main factors underlying their existence, is important both to understand boreal forest dynamics, and to project future changes of natural vegetation cover and their effect on climate.

We do acknowledge that vegetation and climatic variables are linked through a more differentiated set of interactions than just mean annual rainfall, temperature, and forest cover. Henceforth, to improve our understanding of the boreal ecosystem dynamics, we investigate the impact of eight globally-observed environmental variables (EVs) on the tree cover fraction (TCF) distribution. To do so, we make use of satellite products spanning the time period up until 2010, incorporating both spatial and temporal information in our analysis, and taking into account the past variability of the boreal ecosystem. Furthermore, we investigate whether the three observed vegetation modes could represent alternative stable tree cover states. To such avail, we adoperate generalised additive models (GAMs), conditional histograms, phase-space analysis, and statistical tests.

In a similar fashion, it has previously been hypothesised that tropical forests and savannas can be alternative stable states under the same environmental conditions. Evidence for bistability in the tropics has been inferred through fire exclusion experiments (Moreira, 2000; Higgins et al., 2007), field observations and pollen records (Warman and Moles, 2009; Favier et al., 2012; de L. Dantas et al., 2013; Fletcher et al., 2014), mathematical models (Staver and Levin, 2012; van Nes et al., 2014; Baudena et al., 2014; Staal et al., 2015), and satellite remote sensing (Hirota et al., 2011; Staver et al., 2011a, b; Yin et al., 2015).

One key evidence is that the tree cover distribution in the tropics is trimodal (Hirota et al., 2011). In fact, multimodality of the frequency distribution can be caused by the existence of alternative stable states in the system (Scheffer and Carpenter, 2003). In the case of the tropics, multimodality could be an artefact of satellite data processing (Hanan et al., 2014), however, it has been suggested that this issue is not of major importance (Staver and Hansen, 2015). The proposed mechanism responsible for the forest-savanna bistability is a positive feedback between tree cover and fire frequency. The same mechanism has also been employed to explore the potential of multiple stable states in a global dynamic vegetation model (Lasslop et al., 2016). Per contra, it has been suggested that trimodality of the tree cover distribution is not necessarily due to wildfires, since it can be achieved through nonlinearities in vegetation dynamics and strong climate control (Good et al., 2016). The picture is far from complete, as there is evidence that other environmental factors might play a fundamental role in controlling the tree cover distribution (Mills et al., 2013; Veenendaal et al., 2015; Staal and Flores, 2015; Lloyd and Veenendaal, 2016).

## 2 Methods and Materials

### 2.1 Environmental Variables

We study the link between the tree cover fraction distribution of eight globally-observed environmental variables (EVs): mean annual rainfall (MAR), mean minimum temperature (MTmin), growing degree days above $0°C$ (GDD0), permafrost distribution (PZI), mean spring soil moisture (MSSM), wildfire occurrence frequency (FF), soil texture (ST), and mean thawing depth (MTD). These factors are chosen based on the work of Kenneth Hare and Ritchie (1972), Woodward (1987), Bonan (1989), Bonan and Shugart (1989), Shugart et al. (1992), and Kenkel et al. (1997), as they represent the main drivers of the boreal forest biome. Environmental variables can be broadly grouped into temperature, water availability, and disturbances factors.

Temperature factors include mean minimum temperature, growing degree days above $0°C$, permafrost distribution, and mean thawing depth. Soil and air temperature are two major factors responsible for boreal forest structure and dynamics (Kenneth Hare and Ritchie, 1972; Bonan, 1989; Havranek and Tranquillini, 1995). To survive frost and dessication, during winter, coniferous trees enter a period of dormancy, characterised by the suspension of growth processes and a reduction of metabolic activity (Havranek and Tranquillini, 1995). Hence, tree growth and expansion is only possible during extended periods with air temperature above $0°C$. We use growing degree days above $0°C$, calculated from the NCEP/NCAR Reanalysis 1998–2010 (Kalnay et al., 1996), as a measure of the extent of the growing season. Growing degree days above $0°C$ [$°C \, yr^{-1}$], in fact, measure heat accumulation as the sum of the mean daily temperatures above $0°C$ through a year. Furthermore, low soil and air temperatures have several important other consequences. Cold air temperatures are the main regulator of the distribution of permafrost, the condition of soil when its temperature remains below $0°C$ continuously for at least two years. Permafrost and low soil temperatures, on the other hand, impede infiltration and regulate the release of water from the seasonal melting of the active soil layer, inhibit water uptake and root elongation, restrict nutrient availability, and slow down organic matter decomposition (Woodward, 1987; Bonan, 1989). To include these effects, we use the mean minimum temperature at $2 \, m$ [$°C$], and the permafrost distribution [unitless]. Minimum temperatures are obtained from the NCEP/NCAR Reanalysis 1998–2010 (Kalnay et al., 1996). Permafrost distribution is extracted from the Global Permafrost Zonation Index Map (Gruber, 2012), which shows to what degree permafrost exists only in the most favourable conditions or nearly everywhere.

Water availability factors include mean spring soil moisture, mean annual rainfall, and soil texture. In fact, soil moisture and water availability from precipitation are also reflected in the vegetation distribution within the boreal forest biome. Due to permafrost impeding drainage, seasonal snow melt and soil thawing can guarantee a constant supply of water during the growing season (Bonan, 1989). However, this can also cause severe water loss and drought damage when trees are exposed to dry winds or higher temperatures while their roots are still encased in frozen soil and cannot absorb water (Benninghoff, 1952). At the same time, high soil moisture reduces aeration and organic matter decomposition, promoting the formation of bogs, which in turn reduce tree growth and regeneration (Bonan, 1989). To incorporate water importance, we make use of three variables: mean annual rainfall [$mm \, yr^{-1}$] from the CRU TS3.22 1998–2010 dataset (Harris et al., 2014), mean spring soil moisture [$mm$] from the CPC Soil Moisture 1998–2010 dataset (van den Dool et al., 2003), and mean thaw depth [$mm \, yr^{-1}$] from the Arctic EASE-Grid Mean Thaw Depths product (Zhang et al., 2006). Soil water content has also another important

role, as nutrients availability and microbial activities related to nutrient cycling and organic matter decomposition depend on soil water drainage (Skopp et al., 1990). For this reason, we employ soil texture [unitless], from an improved FAO soil type dataset (Hagemann and Stacke, 2014), to describe the type of particles forming it, and to account for nutrients cycling and availability.

Disturbances to vegetation are represented by wildfire frequency. Nutrients cycling, organic matter accumulation, soil moisture, ad soil temperature, are also directly affected by recurring wildfires (Bonan, 1989), which, in addition, change the albedo of the land surface, thus indirectly affecting boreal air temperatures (Flannigan, 2015). Additionally, forest fires can influence the composition and structure of forest communities, as plant species in boreal forests have developed different species-specific traits related to fire occurrences(Rowe and Scotter, 1973; Flannigan, 2015). These adaptations generally allow either to survive

fires, or to promote the establishment of new individuals (Rowe and Scotter, 1973). Different strategies lead to different fire regimes, with implications for climate feedbacks (Flannigan, 2015). Hence, forest fires are a critical component of the boreal forest biome, and we quantify fire frequency [fires $yr^{-1}$] in our analysis using the GFED4 burned area dataset (Giglio et al., 2013), and the Canadian National Fire Database (Canadian Forest Service, 2014). A summary of the variables we use and their origin is presented in Table 1.

To describe tree cover we make use of the percentage tree cover fraction [%] from the MODIS MOD44B V1 C5 2001–2010 product (Townshend et al., 2010). The MODIS tree cover dataset has certain biases and limitations: it underestimates shrubs and small woody plants, as the product was calibrated against trees above 5m tall (Bucini and Hanan, 2007), it never resolves 100% tree cover, it is not well-resolved at low tree cover (Staver and Hansen, 2015), and may not be useful for differentiating over small ranges of tree cover (less than c.10%) (Hansen et al., 2003), as the use of classification and regression trees (CARTs) to

calibrate the dataset might introduce artificial discontinuities (Hanan et al., 2014). Regarding the particular case of the northern latitudes, an evaluation of the accuracy of the MODIS tree cover fraction product pointed out that the dataset may not be suitable for detailed mapping and monitoring of tree cover at its finest resolution (500m per pixel), especially for tree cover below 20%, and that there might be a systematic bias over the Scandinavian region (Montesano et al., 2009). To overcome these limitations, we employ MODIS VCF data at a coarser resolution (0.05°, subsequently re-projected to 0.5°), we aggregate for

most of the analysis tree cover values into three bins encompassing the 0–20, 20–45, 45–100 percent ranges, and we exclude gridcells over Scandinavia from the analysis.

In our analysis, we investigate the use of an alternative dataset for temperatures, namely the CRU TS3.22 tmn product, for the years 1998–2010 (Harris et al., 2014). This dataset has a finer resolution and provides a more detailed picture of the ecosystem, albeit affected by a cold bias over Canada (see CRU TS 3.22 release notes, Harris et al. (2014)). Nonetheless, it shows similar

patterns to the NCEP/NCAR product. The two datasets are heavily linearly correlated, although the CRU tmn product shows lower temperatures, especially over East North Eurasia and East North America. Since our analysis is independent of variables shifts, results obtained using the CRU tmn product are essentially the same (see Supplementary Material).

All datasets are re-projected using CDO (version 1.7.0) on a regular rectangular latitude-longitude grid at 0.5° resolution, and divided into four main areas using approximately the Canadian Shield and the Ural Mountains as middle boundaries for North

America and Eurasia: Western North America (45° N–70° N and 100° W–170° W), Eastern North America (45° N–70° N and

**Table 1.** Variables and datasets summary. Percentage tree cover fraction indicates the proportion of land per gridcell covered by trees. Mean annual rainfall corresponds to the mean cumulative precipitation in mm over a year. Soil moisture is measured as water height equivalents in a 1.6 m soil column. Minimum temperature refers to air temperature at 2 m height. Permafrost zonation index shows the probability of a gridcell to have permafrost existing only in the most favourable conditions or nearly everywhere. Fire frequency is the averaged number of fire events per year. Growing degree days above $0^\circ$C correspond to the sum of the mean daily temperatures at 2 m height above $0^\circ$C through a year, using 6-hourly measurements. Soil texture describes the type of particles forming soil, ranging from sand to clay depending on the particle size. Mean thaw depth corresponds to mm of thawing soil during non-freezing days averaged per year. Surface elevation refers to the topographic altitude per gridcell in m. Land cover type describes the type of vegetation and the density of the cover, independent of geo-climatic zone.

| Variable | Acronym | Units | Origin | Reference |
|---|---|---|---|---|
| Percentage tree cover fraction | TCF | [%] | $0.05^\circ$ MODIS MOD44B V1 C5 2001–2010 product | Townshend et al. (2010) |
| Mean annual rainfall | MAR | [mm yr$^{-1}$] | CRU TS3.22 Precipitation dataset 1998–2010 | Harris et al. (2014) |
| Mean spring soil moisture | MSSM | [mm] | CPC Soil Moisture dataset 1998–2010 | van den Dool et al. (2003) |
| Mean minimum 2m temperature | MTmin | [$^\circ$C] | NCEP/NCAR Reanalysis 1998–2010 | Kalnay et al. (1996) |
| Permafrost zonation index | PZI | [ ] | Global Permafrost Zonation Index Map | Gruber (2012) |
| Fire frequency | FF | [fires yr$^{-1}$] | GFED4 burned area dataset 1996–2012; Canadian National Fire Database 1980–2014 | Giglio et al. (2013); Canadian Forest Service (2014) |
| Growing degree days above $0^\circ$C | GDD0 | [$^\circ$C yr$^{-1}$] | NCEP Reanalysis (NMC initialized) 1998–2010 | Kalnay et al. (1996) |
| Soil texture type | ST | [ ] | improved FAO soil type dataset | Hagemann and Stacke (2014) |
| Mean thaw depth | MTD | [mm yr$^{-1}$] | Arctic EASE-Grid Mean Thaw Depths | Zhang et al. (2006) |
| Surface elevation | | [m] | Global 30-Arc-Second Elevation Dataset | U.S. Geological Survey (1996) |
| Land cover type | | [ ] | Global Land Cover 2000 product (GLC2000) | GLC2000 database (2003) |

$30^\circ$ W–$100^\circ$ W), Western North Eurasia ($50^\circ$ N–$70^\circ$ N and $33^\circ$ E–$68^\circ$ E), and Eastern North Eurasia ($50^\circ$ N–$70^\circ$ N and $68^\circ$ E–$170^\circ$ W). This is done in order to preserve continuity of patterns for the environmental variables and to separate areas with different characteristics, e.g. due to oceanic influence. Note that most of Europe is excluded beforehand due to the high levels of human activity (Hengeveld et al., 2012) and to a possible bias in MODIS data (Montesano et al., 2009). Subsequently, data are filtered to restrict the analysis on areas with minimum anthropogenic influence and where altitude does not play a significant role (Staver et al., 2011b). Areas to exclude are identified using the Global 30-Arc-Second Elevation dataset and the Global Land Cover 2000 product; they correspond to sites that are either bare or flooded (codes: 15 and 19–21), subject to intensive

human activity (codes: 16–18 and 22), or with elevation greater than 1200m. The resulting datasets comprise 5848 gridcells for Eastern North Eurasia (EA_E), 1559 for Western North Eurasia (EA_W), 1775 for Eastern North America (NA_E), and 3094 for Western North America (NA_W).

Within this setup, we assume that the dataset products are suitable for our investigation.

## 2.2 Data Analysis

After filtering and dividing the dataset, we confirm the multimodality of the tree cover distribution in high latitudes, as found by Scheffer et al. (2012) and in line with results from Xu et al. (2015), by optimising the fitting of different sums of Gaussian functions over the tree cover fraction distribution (not shown). Next, we group all data gridcells according to the modal peaks into three states: "treeless", where tree cover is smaller than 20%, "open woodland", with tree cover between 20% and 45%, and "forest", where tree cover is greater than 45%. The ensuing data analysis is aimed at two main purposes: to ascertain the impact of environmental variables on the tree cover, and to assess whether different vegetation states can be found under the same set of environmental variables.

First, we evaluate the link between the eight environmental factors on the tree cover fraction distribution using Generalised Additive Models (Miller et al., 2007). GAMs are data-driven statistical models able to handle non-linear data structures (Hastie and Tibshirani, 1986, 1990; Clark, 2013); their purpose is to ascertain the contributions and roles of the different variables, thus allowing a better understanding of the systems (Guisan et al., 2002). Each GAM test provides an estimate of the proportion of tree cover fraction distribution that can be explained through a smooth of one or more environmental variables (Staver et al., 2011b) - for instance, the formula $TCF = s_1(MTmin) + s_2(MAR)$, with Gaussian family and identity link (see Supplementary Material for further details on the implementation), is used to assess the contribution of minimum temperature and precipitation on the tree cover fraction distribution. For each region, we repeatedly apply GAMs including different combinations of variables, and - to determine whether the sample size influences the results - we use in turn either multiple random samples of 500 gridcells each, multiple random samples of 1000 gridcells each, or all the gridcells.

Subsequently, we analyse the conditional 2-dimensional phase-space between the environmental variables to visualise whether intersections of vegetation states in each phase-space are possible or not. To do so, we perform a kernel density estimation (KDE) of the joint distribution between the two environmental variables, conditioned to whether or not the corresponding data belong to the treeless, open woodland, or forest state, and we plot the KDE together with the environmental variables histograms. Kernel density estimates are used to approximate the probability density function underlying a set of data (Silverman, 1981, 1986).

Next, after excluding growing degree days above $0°C$ and mean thaw depth (see Section 3.2 for details), we look at the 6-dimensional phase-space formed by mean annual rainfall, mean spring soil moisture, mean minimum temperature, permafrost distribution, wildfire frequency, and soil texture, and we divide it into classes in the following manner. First, for every region, we divide the domain of each environmental variable into bins. To do so, we compute the 10th and 90th percentile of the three vegetation states with respect to every environmental variable except soil texture. Then, for the same variables, we select the second lowest 10th and second highest 90th percentiles; these two values are the boundaries of the first and last bin, while the

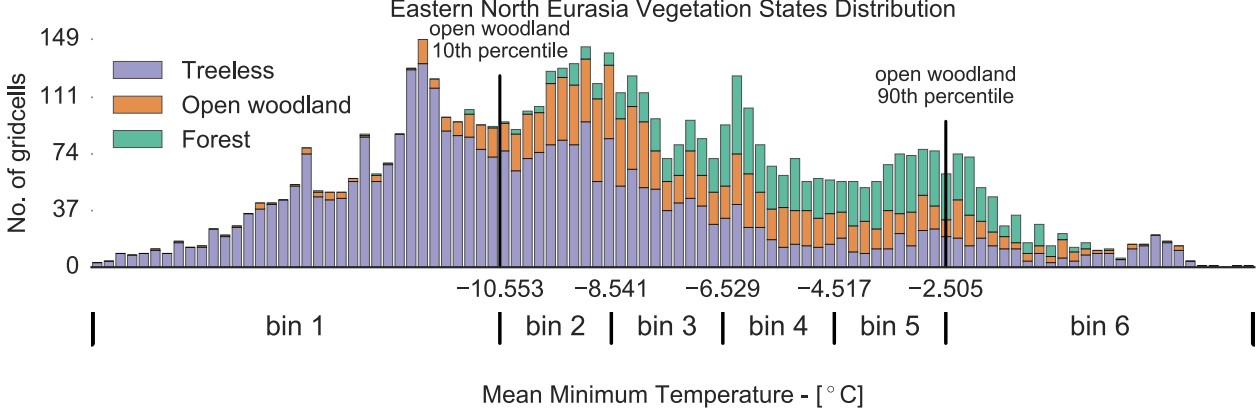

**Figure 1.** Bin division of mean minimum temperature for Eastern North Eurasia. The boundaries of the first and last bins are calculated using the second lowest 10th percentile and second highest 90th percentile of the three vegetation states, with respect to the environmental variable in use, having in mind that only one vegetation state is generally found below or above this thresholds, respectively. The remaining space is subdivided uniformly.

range in between them is equally divided into bins: 5 for MTmin, MSSM, and MAR, and 3 for FF and PZI, as exemplified in Fig. 1 for MTmin; ST is instead divided according to the clay, sand, and loam groups. By doing so, we separate the range of an environmental variable where overlaps between the KDEs of the vegetation states are more likely to happen, from ranges where only one vegetation state is more likely to be found (respectively the central bins and the two most external ones). Second, we consider the partition of the 6D phase-space among the environmental variables generated by the so computed bins. Each element of this partition is defined as a class, i.e., a class is a set of bins for the environmental variables. The idea behind this analysis is to split the 6D environmental variables space into classes where environmental variables could be considered equal for all geographical gridcells. The question, then, is whether the tree cover could be different under the same environmental conditions.

Afterwards, to assess our research question, we associate every geographical gridcell of the boreal area with its vegetation state and with the class corresponding to its environmental variables values. Subsequently, we select two types of areas of interest, that correspond to possible alternative states:

– equivalent tree cover states, defined as gridcells with different vegetation state but same environmental variables class, e.g., an open woodland gridcell and a forest gridcell, where all the environmental variables are in the same bins;

– fire disturbed (FD) tree cover states, defined as gridcells with different vegetation state, where the environmental variables are in the same bins, except for wildfire frequency, e.g., a forest gridcell with low fire frequency and an open woodland gridcell with higher fire frequency but with the remaining environmental variables in the same bins.

Within this last step, to take into account internal variability and the continuous evolution of the ecosystem, we consider only environmental classes that appear significantly, i.e., with a number of gridcells per vegetation state greater than 1% of the total amount of gridcells for that same vegetation state within the entire region (see Supplementary Material for further details). Furthermore, we test whether the tree cover fraction distribution over gridcells with equivalent and fire disturbed tree cover states is multimodal or unimodal. To assess this, we employ the Silverman's test against the hypothesis of unimodality (Silverman, 1981; Hall and York, 2001). Finally, to ascertain that results cannot be explained by the internal variability of the ecosystem alone, we compute the standard deviation of the tree cover fraction distribution for the period 2001–2010 over the same alternative states gridcells, and we compare it with the distributions of the alternative states.

The entire analysis is carried out using Python 2.7.10, IPython 4.0.1, and RStudio 0.99.441.

## 3   Results

### 3.1   GAMs Results

Eastern North America is the region with the highest GAMs results, with more than 80% of the total deviance of tree cover explained, and every variable except fire frequency yielding higher results than in the other three regions. Additionally, the impact of environmental variables on the tree cover fraction distribution depends on the region of interest, as can be seen in Table 2. For instance, soil texture influence ranges from 9–15% to 42–52% in Western and Eastern North America, respectively. A summary of GAMs results using random samples of 1000 gridcells is reported in Table 2.

Growing degree days above $0°C$ and mean minimum temperature are the environmental variables with the greatest influence on the tree cover distribution, with a combined effect ranging from 42 to 77%, in line with literature, as temperature is the main limiting factor for boreal forest (Bonan and Shugart, 1989). The next environmental variable in order of importance is permafrost distribution, with an impact ranging from 10–17% to 69–75% depending on the southern extent of continuous permafrost. Water availability, as expressed through the combined effect of rainfall and soil moisture, explains 26 to 62% of the tree cover distribution. The two variables have a similar influence when considered alone, although MAR has always a greater impact. The impact of wildfires depends heavily on the region of interest, with FF contributing the lowest in Eastern North Eurasia and the highest in Western North Eurasia, 2–9% and 15–20% respectively. Soil related variables, namely soil texture and thaw depth, have a similar impact, generally around 30%.

The environmental variables are not independent of each other, and hence the combined impact of multiple variables does not correspond to the sum of the single terms. For instance, PZI, MTmin, and GDD0, are highly correlated, and their combined effect is only slightly greater than the effect of each factor alone. Overall, the combined effect of all the environmental variables contributes to 52–67% of the tree cover fraction distribution, with the exception of Eastern North America, where the cold temperatures, permafrost distribution, and rainfall gradients, clearly dominate the tree cover distribution and make up for almost 80% of it (omitted from Table 2). We obtain similar results when combining temperature related environmental variables (GDD0, MTmin) with water related ones (MAR, MSSM).

**Table 2.** Summary of GAMs performed using random samples of 1000 gridcells each. The ranges represent the spread of results obtained with different samples, whereas the values in parenthesis correspond to the average from the samples. Statistical $p$-values are $< 0.0001$ for every case. Percentages of explained deviance are a measure of the goodness of fit of each GAM (McCullagh and Nelder, 1989; Agresti, 1996). Reported values are related to the influence on tree cover fraction distribution of mean annual rainfall (MAR), mean minimum temperature (MTmin), growing degree days above $0°C$ (GDD0), permafrost distribution (PZI), mean spring soil moisture (MSSM), wildfire occurrence frequency (FF), soil texture (ST), mean thawing depth (MTD). Values are divided within the four regions of interest, namely, Eastern North Eurasia (EA_E), Western North Eurasia (EA_W), Eastern North America (NA_E), and Western North America (NA_W).

| Variables | Deviance of TCF Explained - % | | | |
| --- | --- | --- | --- | --- |
| | EA_E | EA_W | NA_E | NA_W |
| MAR | 24–30 (27) | 28–38 (32) | 51–57 (55) | 28–36 (32) |
| MSSM | 12–20 (16) | 20–29 (25) | 43–53 (47) | 11–21 (15) |
| MTmin | 36–44 (40) | 23–31 (27) | 70–75 (72) | 36–43 (40) |
| PZI | 38–45 (42) | 10–17 (13) | 69–75 (71) | 31–37 (34) |
| FF | 2–9 (5) | 15–20 (18) | 8–13 (11) | 11–19 (14) |
| GDD0 | 49–57 (54) | 40–51 (46) | 70–74 (71) | 24–34 (28) |
| ST | 9–18 (12) | 26–35 (30) | 42–52 (47) | 9–15 (12) |
| MTD | 21–33 (26) | 27–37 (32) | 39–46 (43) | 18–30 (23) |
| MAR+MSSM | 26–31 (28) | 29–41 (34) | 56–62 (59) | 31–38 (34) |
| MTmin+GDD0 | 53–60 (56) | 43–54 (49) | 73–77 (75) | 42–50 (46) |
| PZI+FF | 42–48 (46) | 34–42 (36) | 70–76 (73) | 34–42 (38) |
| All | 60–67 (63) | 52–58 (55) | 80–85 (82) | 59–65 (62) |

Performing GAMs analysis using all the gridcells or random samples of 1000 gridcells yields similar results, with explained deviances for the former case in between the extremes of the latter, and always with statistical $p$-value$< 0.0001$. On the other hand, using samples of 500 gridcells can increase the explained portion of TCF distribution at the expenses of statistical significance, due to higher $p$-values, and larger-scale applicability. Furthermore, the percentage of explained tree cover fraction distribution is reduced ($\sim 40\%$ maximum combined deviance explained) if we perform the analysis on broader regions than the ones here considered, i.e., on the entire boreal area at once or on the single continents.

## 3.2 Phase-space analysis

Combining together environmental variables in phase-space and performing a kernel density estimation of the joint distribution between the two environmental variables, conditioned to whether or not the corresponding data belong to the treeless, open woodland, or forest state, it is possible to locate peaks in the distributions of the vegetation states.

In many phase-space regions, environmental conditions support only a single "dominant" vegetation state. For instance, low values of GDD0 clearly denote a peak in the distribution of the treeless state. Unfortunately, GDD0 does generally not separate well between the vegetation states in the central area of its distribution, and even combining it with other variables, a clear picture does not emerge. For this reason, and for its high correlation with MTmin (Pearson's correlation coefficient $0.78 < r < 0.94$), GDD0 is not used in the classification. Similarly, MTD is also excluded. Nonetheless, peaks of the KDEs are not always completely disjoint, and it is possible to find intersections between the KDEs of the different vegetation states, as for the case of mean annual rainfall and mean minimum temperature with values around $400 \, \mathrm{mm}$ and $-7 \, °\mathrm{C}$, respectively, where both forest and open woodland are possible. This means that the same environmental conditions can lead to different vegetation states, hinting at possible alternative states.

As a representative case, phase-space plots for Eastern North Eurasia are shown in Fig. 2. Particularly, Fig. 2a represents the KDE of the joint distribution between MAR and MTmin. Each colour is associated with a vegetation state: green for forest, orange for open woodland, and purple for treeless. The isolines describe the probability of finding the three vegetation states under the specific environmental variables regimes, with intense colours indicating higher probabilities. The marginal distributions are reported on the sides of the plot in the form of histograms. The intersections of isolines marked in Fig. 2a show phase-space regions where the same environmental conditions can lead to different vegetation states. Similarly, Fig. 2b represents the KDE of the joint distribution between MSSM and GDD0, with highlighted areas where a single dominant vegetation state is supported by the environmental variables.

Results vary by region, and a complete description of all the combinations between variables is beyond the scope of this paper. Suffices to say that extremes in the distributions of environmental variables are generally associated with a single vegetation state, as in Fig. 2b, whereas intermediate values allow for both single states and intersections, Fig. 2a and 2b, respectively. However, these intersections consider only two environmental variables at a time and they provide only part of the total picture. Results from the classification described and discussed in Sections 2.2 and 3.3 cover all the environmental variables at once.

### 3.3 6D phase-space classification

Associating to every gridcell a class based on the values of the environmental variables reveals that in most cases (2527 classes out of 2546) there is a uniquely determined vegetation state for every class of environmental variables. However, 14 classes allow for different vegetation states, namely either treeless and open woodland, or forest and open woodland. Gridcells belonging to these classes are called equivalent tree cover states. Furthermore, by selecting gridcells corresponding to classes differing only in the fire regime, we can isolate fire disturbed tree cover states, where wildfires played a major role in the timespan covered by the satellite observations (5 classes). A summary of the possible vegetation states found in the system is provided in Table 3, divided into unimodal, multimodal, and fire disturbed states. Equivalent tree cover states gridcells and fire disturbed tree cover states gridcells are represented in Fig. 3 and they cover approximately $\sim 5\%$ of the total boreal area. Specifically, each class contains on average 29 gridcells. Note that we excluded classes containing less than $1\%$ of the gridcells corresponding to each vegetation state. Equivalent tree cover states can be found in every region, with a total of 14 different

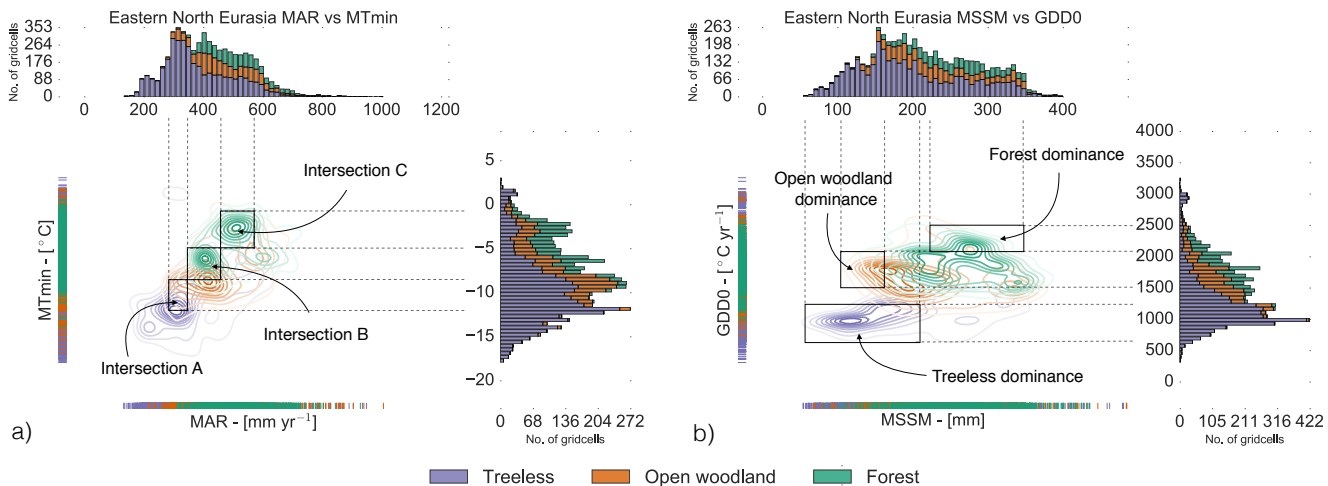

**Figure 2.** Representation of the KDEs of the three vegetation states in the phase-space generated by mean minimum temperature and mean annual rainfall (a), and mean spring soil moisture and growing degree days above $0°C$ (b), for Eastern North Eurasia. Vegetation states are colour-coded as follows: green for forest, orange for open woodland, and purple for treeless. The isolines describe the probability of finding the three vegetation states under the specified environmental variables regimes, with intense colours indicating higher probabilities. Highlighted intersections in phase-space represent areas with different vegetation states under the same environmental conditions (a), whereas the marked areas with only one dominant state hint at the unimodality of the underlying distribution (b). Marginal distributions for the variables are reported to the sides of the plots in the form of histograms.

environmental variables classes related to them, whereas fire disturbed (FD) states appear consistently only in Eastern North Eurasia, and consist of 5 environmental variables classes, of which 4 are also related to equivalent tree cover states. All 19 classes are reported in Table 4. Qualitative indexes for the environmental variables, except for ST and PZI, represent the bin into which the variable's value falls in the classification, as described in Section 2.2; the order is: very low, low, medium-low, medium-high, high, very high. Precise values are reported in Table 4 (see Supplementary Material for further details). Soil texture is described as belonging to the sand, loam, or clay group. Permafrost is described as sparse, discontinuous, frequent, or continuous. Each environmental variable class contains two possible vegetation states, e.g., forest and open woodland, that are consistently found under the same specified environmental regimes.

Table 4 and Figure 3 pinpoint the conditions and locations, respectively, of the possible alternative tree cover states in the boreal area. To test whether the distributions of the possible alternative tree cover states are multimodal, we employ the Silverman's test. Each Silverman's test assesses the hypothesis that the number of modes of the distributions of the alternative open woodland and treeless gridcells, and of the alternative open woodland and forest gridcells, is $\leq 1$. The tests show that the minimum number of modes to describe the distributions is two, for both cases, with $p$-values smaller than $0.001$ and $0.01$, respectively. Figure 4 shows the results of the Silverman's tests on the distributions of the possible alternative tree cover states,

**Table 3.** Summary of possible vegetation states, divided as monostable, bistable, and fire disturbed. Fire disturbed states have a higher fire regime than the indicated counterpart. Treeless always refers to TCF<20%, open woodland to 20%≤TCF<45%, and forest to TCF≥45%.

| Monostable | Bistable | Fire disturbed |
| --- | --- | --- |
| Treeless | Treeless - Open woodland | Open woodland - FD Treeless |
| Open woodland | | Forest - FD Open woodland |
| Forest | Forest - Open woodland | Open woodland - FD Forest |

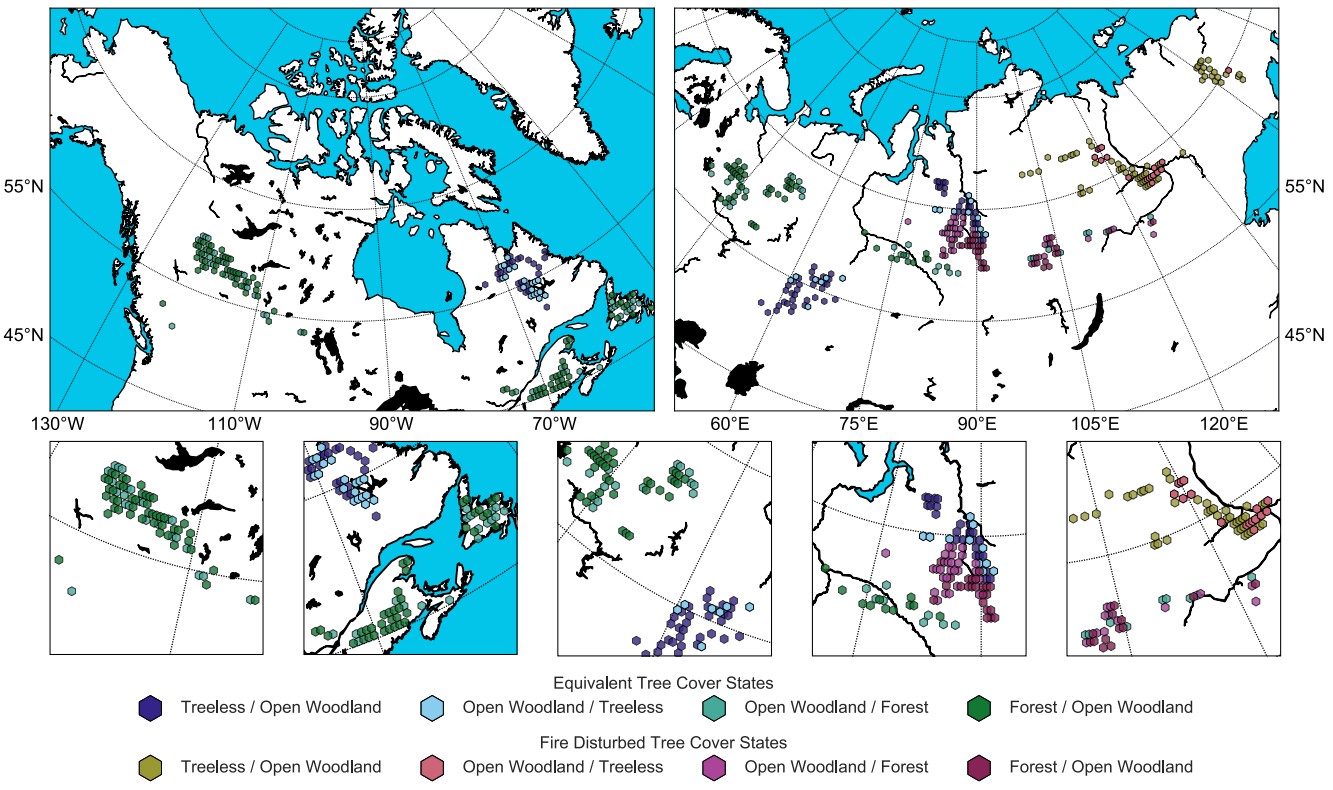

**Figure 3.** Possible alternative tree cover states over North America (left) and North Eurasia (right). The bottom five panels represent a close-up of the areas of interest ordered from West to East. Legend to be interpreted as follows: for every entry in the legend, the first name refers to the observed vegetation state in a specific gridcell, the second name corresponds to the possible alternative state found elsewhere under the same environmental conditions. Fire disturbed tree cover states are only present Eastern North Eurasia.

**Table 4.** Classes related to equivalent tree cover states and fire disturbed (FD) tree cover states. The qualitative marks for fire frequency, mean annual rainfall, mean spring soil moisture, and mean minimum temperature are relative to the extremes of their distributions in the region of interest, and represent the bins into which the phase-space is subdivided. Precise values for these bins are reported in brackets. Soil texture is described as belonging to the sand, loam, or clay group. Permafrost is described as sparse, discontinuous, frequent, or continuous. Each environmental variable class contains two possible vegetation states, e.g., forest and open woodland, that are consistently found under the same specified environmental regimes.

| Region | Case & Vegetation states | FF | ST | PZI | MAR | MSSM | MTmin | Gridcells |
|---|---|---|---|---|---|---|---|---|
| NA_W | 1 Forest - Open Woodland | medium-low [0.29;0.59] | loam | sparse | medium-high [378;471] | medium-low [188;239] | medium-high [−3.6;−1.0] | 27 |
| | 2 Forest - Open Woodland | medium-low [0.29;0.59] | clay | sparse | medium-high [378;471] | medium-low [188;239] | medium-high [−3.6;−1.0] | 44 |
| NA_E | 3 Treeless - Open Woodland | very low [0;0.07] | sand | frequent | low [535;647] | high [427;490] | medium-high [−2.6;−0.65] | 24 |
| | 4 Treeless - Open Woodland | very low [0;0.07] | sand | continuous | very low [0;535] | medium-high [364;427] | medium-low [−4.6;−2.6] | 20 |
| | 5 Forest - Open Woodland | very low [0;0.07] | sand | sparse | very high [984;1607] | very high [490;598] | very high [1.3;5.9] | 58 |
| EA_W | 6 Treeless - Open Woodland | very high [0.59;3.18] | loam | sparse | high [615;663] | very low [99;257] | very low [−8.3;−4.8] | 40 |
| | 7 Forest - Open Woodland | very low [0;0.26] | sand | sparse | medium-high [568;615] | high [327;361] | high [−0.2;2.0] | 18 |
| | 8 Forest - Open Woodland | very low [0;0.26] | loam | sparse | high [615;663] | high [327;361] | medium-low [−4.8;−2.5] | 20 |
| EA_E | 9 Treeless - Open Woodland | very low [0;0.41] | loam | frequent | medium-low [340;468] | very high [332;573] | high [−4.5;−2.5] | 35 |
| | 10 Treeless - Open Woodland | medium-low [0.41;0.82] | loam | continuous | very low [132;331] | low [155;199] | very low [−17.9;−10.5] | 34 |
| | 11 Forest - Open Woodland | very low [0;0.41] | loam | frequent | medium-low [340;468] | medium-low [199;243] | medium-low [−8.5;−6.5] | 23 |
| | 12 Forest - Open Woodland | very low [0;0.41] | loam | frequent | medium-high [468;537] | very high [332;573] | high [−4.5;−2.5] | 23 |
| | 13 Forest - Open Woodland | medium-low [0.41;0.82] | loam | frequent | medium-high [468;537] | very high [332;573] | high [−4.5;−2.5] | 21 |
| | 14 Forest - Open Woodland | very low [0;0.41] | loam | frequent | high [537;606] | high [288;332] | high [−4.5;−2.5] | 19 |
| FD EA_E | 15 Open Woodland - FD Treeless | very low [0;0.41] | loam | continuous | very low [132;331] | low [155;199] | very low [−17.9;−10.5] | 68 |
| | 16 Open Woodland - FD Treeless | medium-low [0.41;0.82] | loam | continuous | very low [132;331] | low [155;199] | very low [−17.9;−10.5] | 35 |
| | 17 Open Woodland - FD Forest | very low [0;0.41] | loam | frequent | medium-high [468;537] | very high [332;573] | high [−4.5;−2.5] | 11 |
| | 18 Forest - FD Open Woodland | very low [0;0.41] | loam | frequent | medium-low [340;468] | medium-low [199;243] | medium-low [−8.5;−6.5] | 11 |
| | 19 Forest - FD Open Woodland | very low [0;0.41] | loam | frequent | medium-high [468;537] | very high [332;573] | high [−4.5;−2.5] | 17 |

confirming their bimodality, together with the respective tree cover distributions. It is clear in Fig. 4 that both cases exhibit a decrease in frequency around 20 and 45 percent tree cover.

Furthermore, we test whether the tree cover modes can be a product of internal variability alone. To do so, we fit the distributions of the possible alternative tree cover states using KDEs, we estimate the distances between the peaks of the distributions, and we compare them with the standard deviation of the tree cover fraction distribution during the 2001–2010 time interval, as a measure of variability. The minimum distance between peaks corresponding to different vegetation states is 18.19 percentage points (note that tree cover fraction is measured as a percentage), whereas the average standard deviation for the alternative states gridcells is 5.77 percentage points, with only one gridcell possessing a variability greater than 18 percentage points. Henceforth, the bimodality of the alternative states distributions cannot be explained by the variability of the tree cover fraction alone. A comparison between the distributions of the alternative tree cover states, the estimated modal peaks, and internal variability is presented in Fig. 5.

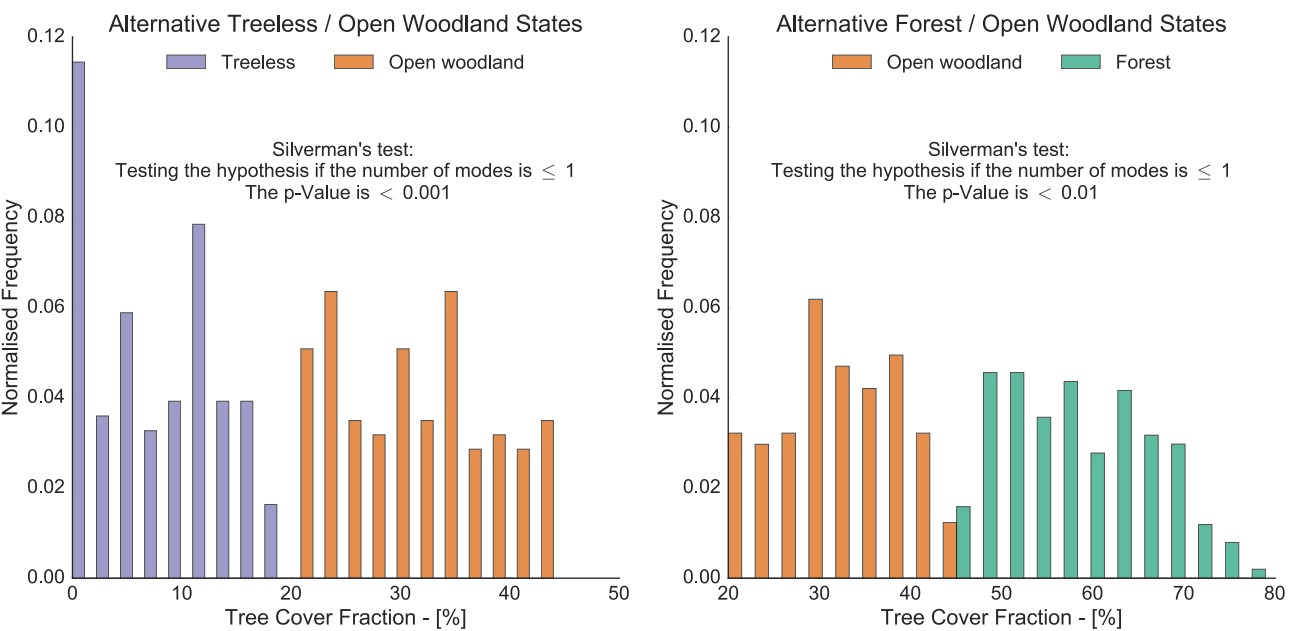

**Figure 4.** Tree cover fraction distribution over the gridcells where equivalent or fire disturbed open woodland and treeless states are found (left), and where equivalent or fire disturbed open woodland and forest states are found (right). For each case the Silverman's test verifies the hypothesis that the distribution is unimodal. The p-value is low in both cases, confirming the multimodality of the distributions.

Notably, equivalent tree cover states generally fall into two categories: either they possess intermediate values for the environmental variables, or they have contrasting ones. For instance, case number 1 in Table 4 is characterised by medium or intermediate values for all the environmental variables, whereas case number 6 shows high values for FF and MAR, but very low for MSSM and MTmin. The first category, with intermediate values, can be associated with transition zones, when passing from an environmental variable class where only a single vegetation state is dominant, to a class where another state is dom-

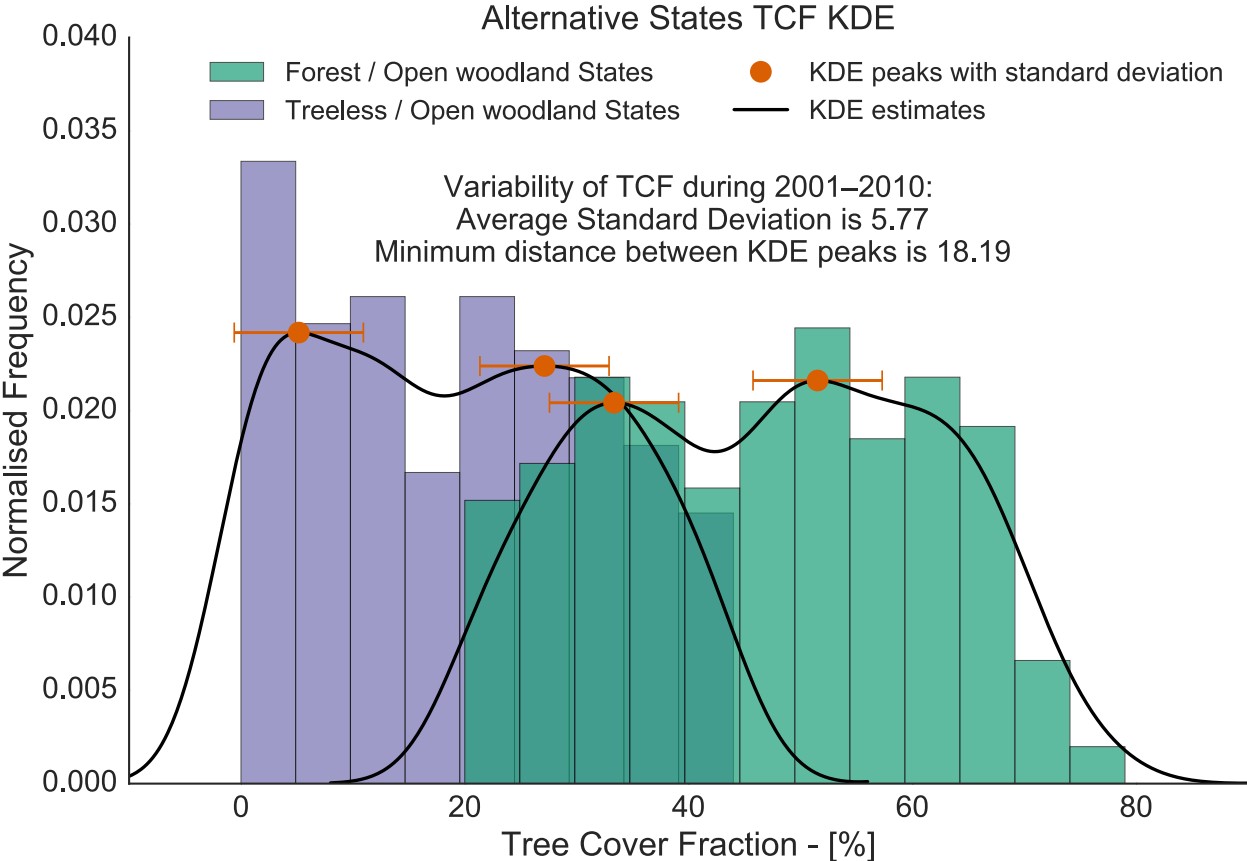

**Figure 5.** The histogram shows the tree cover fraction distributions of the possible alternative tree cover states compared with tree cover fraction internal variability. Purple bars refer to treeless / open woodland states, and green to forest / open woodland states. The black lines and the orange dots represent the kernel density estimate fittings of these distributions and the locations of their modal peaks, respectively. Internal variability of the tree cover fraction distribution for the period 2001–2010, computed as the standard deviation of the distribution, is 5.77 percentage points, and is represented as the orange error-bars. The minimum distance between peaks corresponding to different vegetation states is 18.19 percentage points and is higher than what internal variability could explain.

inant. As a result, the observed tree cover fraction distribution can oscillate between the two states. The second category, on the other hand, relates to classes where at least one of the environmental variables has a value contrasting with the remaining ones. For instance, in case 8, PZI, MAR, MSSM, and ST, all possess values generally associated with forest states, however, MTmin is low, preventing tree growth. This possibly creates a limit cycle where the ecosystem alternates between the different alternative states. Fire disturbed tree cover states, instead, can be grouped into three categories. The first category is represented by classes where the vegetation state with the lowest tree cover is disturbed by fire, and the one with highest tree cover corresponds to one of the existing equivalent tree cover states (case 16, 18, and 19). The second category is the opposite: the

vegetation state with the highest tree cover is disturbed and the one with the lowest tree cover is found among the equivalent tree cover states (case 17). The third category corresponds to the first one, but neither of the vegetation states is found among equivalent tree cover classes (case 15, although very similar to case 10).

Classification results suggest that environmental variables exert a strong, albeit sensitive, control over the tree cover distri-
bution. Depending on the conditions, only one of the three possible vegetation states is attained; for instance, in Eastern North America, classes with very low mean annual rainfall and mean minimum temperature (MAR below $500 \, \text{mm yr}^{-1}$ and MTmin lower than $-9°\text{C}$, see Supplementary Material) are associated with treeless gridcells. In $95\%$ of the gridcells, environmental conditions uniquely determine the vegetation state. However, in transition zones with intermediate or contrasting conditions, it is possible to find multiple vegetation states with the same environmental regimes. In such zones, disturbances could shift
the system between the possible alternative states. In this sense, fire is part of the environment both as a variable (Wirth, 2005; Schulze et al., 2005) and as a disturbance. Strong fire events in transition zones can determine which of two alternative states the system will fall into. On the other hand, changes due to fire events in a stable area should be reabsorbed with time, unless they are so dramatic to produce changes in another main environmental variable, creating a new transition zone.

## 4   Discussion

The link between environmental variables and tree cover fraction varies within the four boreal regions here considered, as described in Section 3.1, and is stronger in Eastern North America, where the cold temperatures, permafrost distribution, and rainfall gradients, dominate the tree cover distribution. Furthermore, the percentage of explained tree cover fraction distribution is greatly reduced when performing the analysis on broader regions, such as the entire boreal area at once or on the single continents. We hypothesise this is caused by the different species distribution across the regions and by the different species-
specific adaptations to the surrounding environment. For instance, North America is mainly dominated by "fire embracing trees", promoting the accumulation of fuel and the occurrence of high-intensity crown fires. On the other hand, Eurasia is populated by "fire resistant trees" in its driest regions, i.e., Eastern North Eurasia, where only surface fires are common, and fire avoiders in Western North Eurasia, which burn less frequently due to the wetter climate of this region (Wirth, 2005; Rogers et al., 2015). As a result, despite the environmental variables having different distributions, the general response of the tree
cover distribution in the four regions is similar, but the impact of each individual environmental variable varies within the regions.

Minimum temperatures and growing degree days are the most influential environmental variables for the boreal tree cover fraction distribution, as can be seen in Table 2. Nonetheless, their combined effect does not fully explain the tree cover distribution, as a more diverse set of variables and feedbacks plays a role. Additionally, the environmental variables are not independent of each other, and hence the combined impact of multiple variables does not correspond to the sum of the single terms. Further-
more, the overall effect of the environmental variables is not able to fully explain the tree cover distribution. We hypothesise this can be linked mainly to three possible causes. First, missing factors in the evaluation, for instance insect outbreaks, which are linked to climate and play an important role in the boreal forest dynamic (Bonan and Shugart, 1989), or grazing from

animals (Wal, 2006; Olofsson et al., 2010). Second, deficiencies in the datasets used, such as the underestimation of fire events in the boreal region (Mangeon et al., 2015), and the limited timespan of satellite observations, as fire return intervals in high latitudes can exceed 200 years (Wirth, 2005). Third, supported by the multimodality of the boreal forest (Scheffer et al., 2012; Xu et al., 2015) and by the results presented in Section 3.3, the presence of areas where the system is in different alternative
stable states under the same environmental conditions.

By linking tree cover distribution to a 6D phase-space formed by environmental variables, we show that under most environmental conditions, the tree cover fraction distribution is uniquely determined, i.e., is unimodal, suggesting a strong control of the vegetation by means of the environment. In this sense, the three different modes of the boreal tree cover distribution (Scheffer et al., 2012; Xu et al., 2015) represent three distinct stable tree cover states that do not generally appear under the
same environmental conditions. However, we find areas where the tree cover fraction distribution is bimodal under the same environmental conditions, suggesting the existence of possible alternative states, as depicted in Fig. 3. These areas are characterised by either intermediate or contrasting environmental conditions, possibly creating limit cycles that allow alternative tree cover states. Furthermore, these areas seem to exhibit a reduced resilience, since disturbances, such as wildfires, appear to be able to shift the vegetation from one state to the other, as in the case of fire disturbed tree cover states. Particularly, Eastern
North Eurasia is the region with the greatest extent of possible alternative tree cover states, and it is the only region where fire disturbed states are found, hinting at a greater susceptibility of its forest resilience.

Environmental conditions control the tree cover distribution in high latitudes, pushing its vegetation towards three distinct tree cover states. This hints at the presence of feedbacks between the vegetation and the environment able to stabilise the vegetation cover in three different ways. However, the environment is influenced by the forest cover state through albedo,
water evapotranspiration (Brovkin et al., 2009), and nutrients recycling. Thus, changes in climate and environmental variables will trigger feedbacks from the vegetation that can either further amplify or dampen the initial changes. In particular, areas of reduced resilience where alternative tree cover states are found, i.e., what we call transition zones, will be affected. As the classification results suggest, environmental variables drive the ecosystem towards seemingly stable states and away from intermediate unstable ones, resulting in the multimodality of the tree cover. Thus, disturbances in transition zones could cause a
rapid ecosystem shift regarding tree cover. Henceforth, it is important to better understand the interplay between environmental variables and tree cover.

Additionally, there are other factors playing a role in the dynamics of the boreal forest, both at local and larger scales. For instance, the understory vegetation acts as an important driver of soil fertility, influencing plant growth and tree seedling establishment (Bonan and Shugart, 1989; Nilsson and Wardle, 2005). An increased nitrogen deposition may promote accumu-
lation of organic matter and carbon in boreal forest (Mäkipää, 1995). At the same time, its effects on the forest floor and soil processes might decrease forest growth (Mäkipää, 1995). Despite its importance, there is a lack of knowledge regarding the impact of understory interactions at large spatial scales, and the contribution of climate change drivers (Nilsson and Wardle, 2005). For these reasons we could not take it into account in our study. Another missing factor is nitrogen (N), as plant growth in the boreal forest is thought to be generally N limited (Mäkipää, 1995). Additionally, herbivore grazing is also influenced
by N fertilisation (Ball et al., 2000), with the potential to affect feedbacks involving soil nutrient cycle and plant regeneration

(Wal, 2006). However, globally-distributed datasets for N availability and grazing pressure suitable for our analysis are not yet available. Local topography also plays a role, as the low solar elevation angle at high latitudes accentuates the effect of ground characteristics such as slope and aspect (Rydén and Kostov, 1980; Bonan and Shugart, 1989; Rieger, 2013), affecting temperature and soil moisture. Finally, micro-topography, such as shelter from boulders, can increase resistance to disturbances by creating small-scale refugia (Schmalholz and Hylander, 2011), thus locally increasing the resilience of the forest.

In the context of climate change, understanding transition zones at large scales is necessary for assessing future projections of vegetation cover. Climate change is impacting the boreal area more rapidly and intensely than other regions on Earth; for instance, surface temperature has been increasing approximately twice as fast as the global average (IPCC, 2013). Temperature is a key variable in this region, as it is connected with tree growth and mortality cycles, with permafrost thawing and the hydrological cycle, and with disturbances, such as wildfires and insect outbreaks (Assessment, 2005; Wolken et al., 2011; Johnstone et al., 2010; Scheffer et al., 2012; D'Orangeville et al., 2016). Particularly, air and surface warming can increase the frequency and extent of severe fires (Flannigan et al., 2005; Balshi et al., 2009; Johnstone et al., 2010), and promote more favourable conditions for insect outbreaks (Volney and Fleming, 2000). At the same time, climate change influences the resilience of boreal forest stands (Johnstone et al., 2010), making them more susceptible to abrupt shifts due to disturbances. As temperature increases and permafrost thaws, it is more likely to find intermediate conditions where alternative tree cover states are possible. For instance, a study on the southern part of the eastern North America boreal forest has shown that an increased disturbance regime, together with the superimposition of fires and defoliating insect outbreaks, can cause a shift between alternative vegetation states (Jasinski and Payette, 2005). Furthermore, there is strong evidence that certain types of extreme events, mostly heatwaves and precipitation extremes, are increasing under the effect of climate change (Orlowsky and Seneviratne, 2012; Coumou and Rahmstorf, 2012). Such events could foster areas with contrasting environmental conditions, further weakening the stability of the boreal ecosystem, and increasing its susceptibility to shifts.

## 5   Conclusions

Through the analysis of generalised additive models, we find that the environment exerts a strong control over the tree cover distribution, forcing it into distinct tree cover states. Nonetheless, the tree cover state is not always uniquely determined by the variables at use. Furthermore, the response of vegetation to the environment varies in the four regions considered: Eastern North America, Western North America, Eastern North Eurasia, and Western North Eurasia.

By means of a classification, we analyse the 6D phase-space formed by mean annual rainfall, mean minimum temperature, permafrost distribution, mean spring soil moisture, wildfire occurrence frequency, and soil texture. We find several environmental conditions under which alternative tree cover states are possible, broadly falling into two categories: with contrasting environmental features, e.g. high rainfall but low temperature, or with intermediate environmental values. In regions under these environmental conditions, the tree cover exhibits a reduced resilience, as it can shift between alternative states if subject to forcing.

As fires can shift the tree cover from one vegetation state to another in regions of reduced resilience, we find support for the hypothesis that a strong fire disturbance could permanently change the state of the ecosystem, by the combined effect of a shift in tree cover and its potential feedbacks on the environment.

Finally, we find that regions with possible alternative tree cover states encompass only a small percentage of the boreal area (∼5%). However, since temperature and temperature-related environmental variables exert the strongest control on the tree cover distribution and its modes, temperature changes could greatly affect forest resilience and cause an expansion of regions with alternative tree cover states.

In the context of climate change, a gradual expansion of transition zones with reduced resilience could lead to regional ecosystems shifts with a significant impact not only on the structure and functioning of the boreal forest, but also on its climate.

## 6 Data availability

All data necessary to reproduce the paper are provided in the Supplementary Material.

*Author contributions.* Both authors designed the research; B. Abis performed the research; both authors contributed to the discussion and interpretation of the results; B. Abis wrote the first draft of the manuscrip and both authors contributed to its final draft.

*Acknowledgements.* We would like to thank the Integrated Climate Data Center (ICDC, icdc.cen.uni-hamburg.de) of University of Hamburg for helping with the retrieval of datasets. We thankfully acknowledge Gitta Lasslop for helping us review the manuscript, and Fabio Cresto Aleina for his useful comments. We also thank the International Max Planck Research School on Earth System Modelling (IMPRS-ESM) for supporting this work. Finally, we thank our editor A. Ito and three anonymous referees for their constructive comments on the manuscript.

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
