# Peer review of "Environmental Conditions for Alternative Tree Cover States in High Latitudes"

_Biogeosciences, 2016_

## Referee Comment (RC1) · Anonymous Referee #1 · 22 Nov 2016

**GENERAL COMMENTS**

The manuscript by Abis and Brovkin aims to determine the impact of several environmental variables (EVs) on the tree cover fraction in the boreal areas, and to assess the existence of different vegetation states under the same environmental conditions. Through the combined use of different statistics, they found that regions with same environmental conditions and different tree cover states cover about 5% of the boreal area, whereas in the rest of the regions the EVs determine a single vegetation states. Multimodality in tree cover is largely being studied in the tropical regions, and less works exist about multimodality in boreal forest.

- Overall, this study is interesting and novel. However, I think that, especially in the

Introduction Section, it is not sufficiently framed in the context of cited references regarding the boreal forest biome. In this regard, I suggest to expand the description of the current knowledge about the boreal forest biome.

- Overall, the scientific approach and the applied methods are valid and good, however I think that the links between the different statistics involved should be better described. For example, it should be better clarify in the text how the information learned by applying one statistic are useful for making decision in applying the others.

- Finally, I suggest to revise the structure and the text of the paper in order to avoid repetitions and make the manuscript more readable. In this regard, in particular I suggest to merge the Discussion and Conclusion sections.

In these regards and apart from these, in the following I report more specific comments:

SPECIFIC COMMENTS

- 1 Introduction

Pag 2, Lines 6-10: "To such avail ..[] (Reyer et al. 2015)" I suggest introducing the boreal forest biome before this sentence or immediately after.

In general, I suggest to reduce the description of the bimodality in tropical vegetation and to expand the description of the boreal forest, because at the moment in the manuscript they have almost the same importance. Since the study is about the boreal forest I think that the Introduction should be focused mainly on the state of the art of the study about of this biome, in order also to highlight the novelty of this study.

In particular: Pag. 2, Lines 29-33: Since most of these environmental variables are those considered in this work, could you give more information about their role in the boreal forest biome, and also more information about the cited studies? E.g. if these papers considered only some specific areas, the main knowledge about the variable interactions... Pag. 2 Lines 33-34: In order to explain better the role of boreal forest in the climate system, could you provide a more extended description of some feedbacks

between each other?

Pag. 3 lines 3-7: I suggest to describe more extensively the main outcomes of Scheffer et al. 2012, in particular detailing what they found about the existence of multiple states under the same environmental variables in boreal forest, in order to better introduce the current knowledge about multimodality, what is missing and thus the timely of the study reported in this manuscript.

- 2 Methods and Materials

- 2.1 Environmanetal Variables Datasets

Pag. 3 Lines 20-21: More information about current knowledge on the importance of these variable in the boreal forest biome, which I suggested to include in the Introduction, could alternatively be reported here.

Pag. 3 Line 21: "[. . .] they are summarised in Table 1". I suggest to modify the sentence for inserting also the kind of information provided by Table 1.

Pag. 3 Line 22: Please insert at the beginning of the sentences the name of the tree cover dataset (i.e. MODIS).

Although Table 1 has the references of all the datasets, I suggest to provide, in the text or in an additional column of Table 1, a brief description of all the variables, or at least of the variables that need a definition (e.g. what the permafrost index indicates, the type of soil texture, the definition of GDD0, the depth at which the soil moisture refer to. . .).

- 2.2 Data Analysis

Could you report in the text the information about the GAM implementation? Such as the assumed error distribution of the data and the implemented link function.

Pag. 5, Line 22: why do you use only 6 variables for the multiple-dimensional phase-space instead of 8? This is currently explained later on, but to improve clarity I suggest

to explain the reason here, or to refer directly to Sec. 3.2 for the explanation.

- 3 Results

- 3.3 6D phase-space classification results

How many are the total found classes?

Pag. 10 Lines 1-2: "Qualitative [. . .] high." Is it possible to provide the quantitative values of the extremes of the qualitative index for each variable? For example, what are the extreme values of the qualitative range called "medium-low" for FF? Furthermore, how do these ranges change for the different regions?

Pag. 13. Line 4: "Depending on the conditions, only one of the three possible vegetation states is attained." It is possible to provide some examples?

- Table 1

I suggest including in the table also the measure units of the variables.

- Table 4

I suggest adding a column with the number of gridcells found in each class.

TECHNICAL CORRECTIONS

- 2.2 Data Analysis

Pag. 5, Line 19: "or" instead of "ot", please fix the typo.

Pag. 6, Line 10: Please rephrase the sentence "(generally at least 1% of the gridcells with the same vegetation state)"

- Table 1

Please replace "0.05° MODIS MOD44B V1 C5 2010 product" with "0.05° MODIS MOD44B V1 C5 2001-2010 product"

---

## Referee Comment (RC2) · Anonymous Referee #2 · 22 Nov 2016

This paper covers a very interesting and important topic. Since we can measure tree cover from remote sensing, several papers have shown potential bistability in forest cover. It is of major importance to find environmental conditions which can trigger a shift from one state to another. This paper aims to link the tree cover distribution to observed environmental conditions and thereby conclude that 5% of the boreal area is potentially bistable.

I found the paper difficult to read, mainly due to the many abbreviations. I think that the authors could delete loads of them and just write down the whole names. Further my major concern is the structure of the results. It is unclear what is expected, some parts are discussion already, while some crucial results are not introduced. The authors

should take more time to present their results. I have a set of minor and more major points

Abstract: 1) Please try to minimize the abbreviations. Is it needed to mention them already in the abstract? 2) The aim it to study the impact on the tree cover fraction by eight environmental factors. I think you do not prove that it is the impact; you only link them following a statistical approach? So I would be in favour to change the aim;

Methods 3) It is unclear to me why the authors didn't use a seasonal variable in here. I think that seasonality in the temperature and rainfall will probably tell more than the averages and minimum values. 4) What is permafrost distribution? I am not a specialist on this, but it would be helpful to add more information on the selected environmental variables and also add units in table 1 5) Pg 3, line 30: How many RS-cells is 0.05degree? 6) Also add all abbreviations in table 1 7) Pg 4, line 3: Of course both data sets are highly correlated, but more interesting is to see the anomalies; 8) Pg 5, line 13: just call EV environmental variables. These changes will highly improve the reading 9) Pg 6, line 1, we associate every grid cell? Which grid cell is this the 0.05 degree or the RS-grid cell?

Results 10) Pg 7, Line 3: Here you start referring to the table by not interpreting what we see but only saying that we have uncertainty bands. I think that you first need to introduce what we see highest explained variance is found for NA_E. (In the text this is mentioned as NAE.. please just use the whole name, you also do this later on with Eastern North Eurasia for instance). And that this differs per region etc. Then you should also make a column with average values for all data. A question I do have is if the differences in explaining variance per region are dependent on the range of the environmental variables. With a larger range you would expect a large explaining variance. 11) Results-vs Discussion: I realize that the above section should not be too much discussion. From pg7, L13-15 to pg8 L1-20 onwards you have a mix of results and discussion. These should be separated and should have a new section in the discussion chapter. 12) Phase-space results: Again take more time to introduce Fig2a.

Is it not only the phase-space but also the KDE? Also mention in the text what these intersections are. Mention what the colours are and how we need to interpret Fig2a. After that introduce Fig2b. 13) I also like to see a correlation matrix how the different EV's are correlated with each other. It is now unclear why you show Eastern North Eurasia with the combination of the two and why not a combination of other variables. I think that MAR and mean_TMIN are highly correlated as they are placed on one line, meaning overlap of information. 14) 3.3: I do not understand the part at page9 L5-10. I can see that you are interested in grid cells having similar EV's but not similar tree cover. However I am confused how to read table 3, why is that you have four columns? If you mention a number of classes (page9, L8), what do you mean? 15) It is very interesting to see how these data are clustered. I have problems with reading the different colours in the legend. Also some symbols have a black line and others not, but unclear if this relates to the fire or non-fire disturbed states or does it relate to single stable vs bistable data points?. Can you also see some spatial patterns of data which have the same bistability, but now currently in a different mode? 16) Figure 4: It is Silverman's test (to words) 17) Treeless state: I agree with your statement that tree cover below 20% is difficult to measure wit RS. Therefore I have my doubts about the results of Fig.4 Why is it that you use in that detail the tree cover fractions below 20%? What do you want to show with these figures. There is not much text about figure 4, so is it needed or can you directly introduce figure 5. Although for this figure, the same holds for the data <20%. Discussion 18) Do not understand your statement on pg14, L6; what kind of feedbacks? You didn't study this, so why is it that they might or might not play a role? 19) I found the discussion on N-cycling, decomposition, fertilisation a bit too much detail in comparison to the work you have presented. You have now linked it to soil type, and if you would be more interested in Nitrogen then you could have used modelled maps from DGVM's (as LPJ-Guess) or use maps or soil organic matter. I think that there are more important things to discuss, for instance why individual set of EV's are different between the regions, why fire is that important, which regions are more sensitive to a change in temperature then others, or a change

in permafrost depth etc. etc. So keep the discussion more related to your own findings.

---

## Referee Comment (RC3) · Anonymous Referee #3 · 23 Nov 2016

Comments on BG-2016-401 Title: Environmental Conditions for Alternative Tree Cover States in High Latitudes Author(s): Beniamino Abis and Victor Brovkin

The paper analyzes the impact of eight environmental variables (EVs) on tree cover fraction distribution (TCF) using generalized additive models, conditional histograms and phase-space analysis. Authors found that minimum temperatures and growing degree days were the most influential environmental variables for the boreal TCF distribution. The relationships between tree cover and EVs were different within the four boreal regions. Generally, the paper is well written and addresses relevant scientific questions within the scope of Biogeosciences, and could contribute to the understandings of species distribution in boreal region in response to climate change. However, I

have few minor comments as listed below.

1. Please add citations to the sentence '. . .and to ecosystems exhibiting potential alternative tree cover states under the same environmental conditions. . .' on Page 2 Line 5. 2. Table 2,4 contains many acronyms without complete name in the table caption. The figures or tables should stand alone. Suggest to add complete names to table captions or to the table. What do you mean by 'very low, medium low, very high and medium high' in Table 4? Please quantify or specify them. 3. The results or findings in abstract are too general, e.g. "we find that environmental conditions exert a strong control .. ...", "we find that the relationship between tree cover and environment is different within the four boreal regions. . ..", and etc. Please be specific. 4. Conclusions (line 2-29 on page 15) are too long, more like summary. Please shorten this section.

---

## Author Response (AR1)

**Final Response to the Associate Editor**

B. Abis and V. Brovkin

Dear Editor,

Thank you for the thorough attention dedicated to our manuscript submission. Please, have a look at this document, in which we will provide the replies to the three anonymous referees' reviews, including point-by-point answers to their comments and questions, highlighting line numbers of revisions according to specific comments. Note that we added a few more details to the answer A14 for the C14 query of Referee #1. Furthermore, we will provide a marked-up version of the revised manuscript, with discarded text marked in red colour and additional text in blue. Finally, we would like to express our appreciations to you and to the referees. Thank you very much for your contribution to the manuscript.

Best regards,
B. Abis and V. Brovkin

**Reply to Anonymous Referee #1**

B. Abis and V. Brovkin

Dear Referee,

Thank you for your positively constructive review and for your insightful comments. In this document, we will provide an answer to your comments and queries. We will not make a distinction between your general, specific, and technical comments.

Best regards,
B. Abis and V. Brovkin

C1. "Overall, this study is interesting and novel. However, I think that, especially in the Introduction Section, it is not sufficiently framed in the context of cited references regarding the boreal forest biome. In this regard, I suggest to expand the description of the current knowledge about the boreal forest biome."

A1. Thank you for your comment. We agree that to better frame our work in the context of the boreal forest biome, more information could be beneficial. Following your suggestion, we will expand and restructure the Introduction, to make the boreal forest description more prominent, including details regarding the main feedbacks, and a more detailed explanation of the findings of Scheffer et al. 2012.

C2. "Overall, the scientific approach and the applied methods are valid and good, however I think that the links between the different statistics involved should be better described. For example, it should be better clarify in the text how the information learned by applying one statistic are useful for making decision in applying the others."

A2. From this and other comments, we understood that our explanation of the analysis performed is not straightforward to follow, especially with regards to the flow of decisions and results. Hence, we agree on providing further clarifications and details in the text, to better guide the reader.

C3. "Finally, I suggest to revise the structure and the text of the paper in order to avoid repetitions and make the manuscript more readable. In this regard, in particular I suggest to merge the Discussion and Conclusion sections."

A3. Having evaluated all the comments received, we agree to restructure part of the paper to make it more readable. To this avail, following the comments, we will introduce major modifications to the following sections: Introduction, GAMs Results, Discussion, and Conclusions.

C4. "[Page 2, lines 6–10]: "To such avail... [] (Reyer et al. 2015)" I suggest introducing the boreal forest biome before this sentence or immediately after."

A4. Following your comment, we decided to restructure the Introduction. We will introduce the boreal forest biome after this sentence and expand its description with further details to better frame it in the research context. [Page 2, lines 14–35; page 3, lines 1–7]

C5. "In general, I suggest to reduce the description of the bimodality in tropical vegetation and to expand the description of the boreal forest, because at the moment in the manuscript they have almost the same importance. Since the study is about the boreal forest I think that the Introduction should be focused mainly on the state of the art of the study about of this biome, in order also to highlight the novelty of this study."

A5. Dear Referee, the topic of multistability in tropical vegetation is currently an important hotspot of discussion, with debates over several different aspects of the savanna-forest transitions. Such discussion influenced the way we structured and performed our work. For these reasons, we think that an overview of the discussion ought to be mentioned and that it is not possible to reduce its description. However, we do agree on improving the balance of the Introduction in favour of the boreal forest and on the state of the art of the study of this biome. [Page 2, lines 14–35; page 3, lines 1–6 and 15–30]

C6. "In particular: [page 2, lines 29–33]. Since most of these environmental variables are those considered in this work, could you give more information about their role in the boreal forest biome, and also more information about the cited studies? E.g. if these papers considered only some specific areas, the main knowledge about the variable interactions..."

A6. We agree that to improve the comprehension of the paper, we should provide more information about this environmental variables. Following your second suggestion (comment 9), we will include a paragraph in the Environmental Variables Datasets section about the role and the importance of the environmental variables we used. [Page 4, lines 3–34; page 5, lines 1–16]

C7. "[Page 2, lines 33–34]: In order to explain better the role of boreal forest in the climate system, could you provide a more extended description of some feedbacks between each other?"

A7. We will include a more extended description of the main feedbacks playing a role, so that we can also refer to them more clearly in the Discussion section. Particularly, we will introduce the way boreal forests influence climate through albedo, evapotranspiration, and carbon sequestration. [Page 2, lines 21–29]

C8. "[Page 3, lines 3–7]: I suggest to describe more extensively the main outcomes of Scheffer et al. 2012, in particular detailing what they found about the existence of multiple states under the same environmental variables in boreal forest, in order to better introduce the current knowledge about multimodality, what is missing and thus the timely of the study reported in this manuscript."

A8. Following your comment, we will provide a more thorough description of Scheffer et al. 2012 findings, as they represent the base for our study. [Page 2, lines 34–35; page 3, lines 1–5]

C9. "[Page 3, lines 20–21]: More information about current knowledge on the importance of these variable in the boreal forest biome, which I suggested to include in the Introduction, could alternatively be reported here."

A9. As stated in the answer to comment 6, we will report here more information about the current knowledge on the boreal forest biome, and on on the role and importance of the variables used. [Page 4, lines 3–34; page 5, lines 1–16]

C10. "[Page 3, line 21]: "[. . . ] they are summarised in Table 1". I suggest to modify the sentence for inserting also the kind of information provided by Table 1."

A10. We agree on providing within the text a description of the variables and hence on the information contained in the Table 1. [Page 4, lines 3–34; page 5, lines 1–16]

C11. "[Page 3, line 22]: Please insert at the beginning of the sentences the name of the tree cover dataset (i.e. MODIS). Although Table 1 has the references of all the datasets, I suggest to provide, in the text or in an additional column of Table 1, a brief description of all the variables, or at least of the variables that need a definition (e.g. what the permafrost index indicates, the type of soil texture, the definition of GDD0, the depth at which the soil moisture refer to. . . )."

A11. We will implement your suggestion and we will provide information about all the variables and their role in the boreal forest biome within the text. Furthermore, we will add a more detailed caption for Table 1. [Page 5, line 16; page 6, Table 1]

C12. "Could you report in the text the information about the GAM implementation? Such as the assumed error distribution of the data and the implemented link function."

A12. Dear referee, we think that full details of the GAM implementation will not contribute to improve the paper, as they will make it more technical and harder to read. However, we will mention, as you ask, the family and link function used in our analysis through a suite available on R. Furthermore, additional details regarding not only GAMs, but our entire setup, including all the packages and scripts used, are already present as supplementary material. [Page 7, lines 18–19]

C13. "[Page 5, line 22]: why do you use only 6 variables for the multiple-dimensional phasespace instead of 8? This is currently explained later on, but to improve clarity I suggest to explain the reason here, or to refer directly to Sec. 3.2 for the explanation."

A13. We agree with you and we will include a reference to Section 3.2 for improved clarity. [Page 7, line 29]

C14. "How many are the total found classes?"

A14. Dear Referee, the number of found classes is as follows: 1185 in Eastern North Eurasia, 438 in Western North Eurasia, 457 in Eastern North America, and 835 in Western North America. However, the total amount of unique found classes is not 2915 but 2546. Of these, 19 are multistable or fire disturbed. We will include this information in the text. [Page 11, lines 25–27 and 30–31]

C15. "[Page 10, lines 1–2] "Qualitative [. . . ] high." Is it possible to provide the quantitative values of the extremes of the qualitative index for each variable? For example, what are the extreme values of the qualitative range called medium-low for FF? Furthermore, how do these ranges change for the different regions?"

A15. To make Table 4 more readable, we initially decided to include the information you ask in the supplementary material only. However, we will change this and make Table 4 more complete, with all the ranges and the number of gridcells per class. [Page 12, line 5; page 14, Table 4]

C16. "[Page 13, line 4]: "Depending on the conditions, only one of the three possible vegetation states is attained." It is possible to provide some examples?"

A16. Essentially, in 95% of the cases, the class uniquely determines the vegetation state (either treeless, open woodland, or forest). Hence, we will easily provide an example for each case. [Page 17, lines 5–8]

C17. "[Table 1] I suggest including in the table also the measure units of the variables."

A17. We agree that this will improve the information conveyed by Table 1, hence, we will include units and a more detailed caption. [Page 6, Table 1]

C18. "[Table 4] I suggest adding a column with the number of gridcells found in each class."

A18. We agree that this will add an important information to Table 4, hence we will add such a column. [Page 14, Table 4]

C19. "[Page 5, line 19]: "or" instead of "ot", please fix the typo."

A19. We beg your pardon for the typo; we will correct it immediately. Thanks for noticing. [Page 7, line 25]

C20. "[Page 6, line 10]: Please rephrase the sentence "(generally at least 1% of the gridcells with the same vegetation state)"."

A20. Following your comment, we realised this sentence was somewhat vague. We will rephrased it in a more clear and concise way. [Page 9, lines 2–4]

C21. "[Table 1] Please replace "0.05° MODIS MOD44B V1 C5 2010 product" with "0.05° MODIS MOD44B V1 C5 2001-2010 product"."

A21. We will implement your suggestion. [Page 6, Table 1]

**Reply to Anonymous Referee #2**

B. Abis and V. Brovkin

Dear Referee,

Thank you for your time and for your valuable comments and suggestions. In this document, we will provide an answer to your comments and queries, highlighting how we will modify the manuscript in view of your suggestions.

Best regards,
B. Abis and V. Brovkin

C1. "I found the paper difficult to read, mainly due to the many abbreviations. I think that the authors could delete loads of them and just write down the whole names. Further my major concern is the structure of the results. It is unclear what is expected, some parts are discussion already, while some crucial results are not introduced. The authors should take more time to present their results."

A1. Thank you for your valuable opinion. We thought we would simplify the paper by introducing some abbreviations. However, from your comments it seems like we actually made it harder to read. We will try to reduce the number of abbreviations and write plain sentences when possible. Regarding the structure of the results, we found it difficult to separate some results from their interpretation and discussion. However, following your suggestion, we will restructure them. We will expand the Discussion section and simplify the results one, especially the GAMs Results section, taking more time to introduce the results and moving all the discussion and interpretations in the Discussion section.

C2. "Please try to minimize the abbreviations. Is it needed to mention them already in the abstract?"

A2. We understood it is necessary to introduce abbreviations in the abstract from BG manuscript regulations. However, following your suggestion, we will reformulate the abstract so that it will not make use of them. [Page 1, lines 9–14]

C3. "The aim it to study the impact on the tree cover fraction by eight environmental factors. I think you do not prove that it is the impact; you only link them following a statistical approach? So I would be in favour to change the aim."

A3. Dear referee, the meaning of that particular sentence is that we want to quantify the impact on tree cover, since the primary role that environmental variables exert on the vegetation has already been studied by many before us. To avoid ambiguity, we will follow your suggestion and rephrase our aim so that it is clear that we study the link between the various distributions. [Page 1, line 5]

C4. "It is unclear to me why the authors didnt use a seasonal variable in here. I think that seasonality in the temperature and rainfall will probably tell more than the averages and minimum values."

A4. Dear Referee, you are right in saying that seasonal variables play an important role in the boreal forest dynamics. For this reason, we actually included several indicators that account for seasonality. In particular, the spring soil moisture measures water availability during the thawing period, when plants have access to a deeper active layer and can start to use unfrozen water, whereas the growing degree days above $0°C$ are a proxy for the extent and intensity of the plant growing season. In fact, growing degree days are a measure of heat accumulation, and many developmental events of plants depend on it. Hence, by using degree days above $0°C$ it is possible to estimate the influence of the growing season regardless of differences in temperatures from year to year. On the other hand, we agree that monthly data would provide a finer representation of the different seasonal aspects, however, due to the already high number of variables, such analysis would increase too much the degrees of complexity of the problem, going beyond the scope of the paper. Nonetheless, we recognise that the lack of details about the datasets used and the role of the environmental variables in the boreal forest biome makes it harder for the reader to understand our motivations. For these reasons, we decided to include in the manuscript information about the definition, role, and importance of the variables used for the analysis. [Page 4, lines 3–34; page 5, lines 1–16; page 6, Table 1]

C5. "What is permafrost distribution? I am not a specialist on this, but it would be helpful to add more information on the selected environmental variables and also add units in Table 1."

A5. We agree that units are necessary and we will add them to Table 1. Furthermore, as stated in the answer to comment 4, we will include detailed information on all the variables used. Regarding your question on permafrost, the zonation index shows to what degree permafrost exists only in the most favourable conditions or nearly everywhere. [Page 4, lines 3–34; page 5, lines 1–16; page 6, Table 1]

C6. [Page 3, line 30]: "How many RS-cells is 0.05degree?"

A6. Dear Referee, throughout the entire paper, we make reference to rectangular LONLAT grids. In particular, on a global level, $0.05°$ correspond to a grid with $7200 \times 3600$ griddcells with side length of $\sim5.5$ km. This translates into 1400000 ($2800 \times 500$) griddcells for North America, and 1760000 ($4400 \times 400$) griddcells for Eurasia. The numbers for the $0.5°$ grid are the same divided by 100.

C7. "Also add all abbreviations in Table 1".

A7. We did not understand this comment, as abbreviations are already present in Table 1. We will, however, improve the caption for the table. [Page 6, Table 1]

C8. [Page 4, line 3]: "Of course both data sets are highly correlated, but more interesting is to see the anomalies".

A8. You are right. To deal with this aspect, we made a full analysis of the differences in results due to the use of the two datasets. The findings are already reported in the supplementary material. However, due to the restricted amount of anomalies, the core results regarding transitions zones are essentially the same. [Page 5, line 32]

C9. [Page 5, line 13]: "just call EV environmental variables. These changes will highly improve the reading".

A9. We will implement your suggestion to improve readability and reduce abbreviations.

C10. [Page 6, line 1]: "we associate every grid cell? Which grid cell is this the 0.05 degree or the RS-grid cell?"

A10. Dear referee, in practical terms, there is only one geographical grid used throughout the analysis to which all variables (including tree cover) refer to. It is a rectangular LONLAT grid with $0.5°$ resolution. At every location (what we call gridcell, indicated by its longitude and latitude) we associate the values of all the environmental variables for that specific location and, at this particular step, the value given by the classification.

C11. [Page 7, line 3]: "Here you start referring to the table by not interpreting what we see but only saying that we have uncertainty bands. I think that you first need to introduce what we see highest explained variance is found for NA₋ E. (In the text this is mentioned as NAE. . . please just use the whole name, you also do this later on with Eastern North Eurasia for instance). And that this differs per region etc. Then you should also make a column with average values for all data. A question I do have is if the differences in explaining variance per region are dependent on the range of the environmental variables. With a larger range you would expect a large explaining variance."

A11. Dear Referee, as the answer to this particular comment is somewhat lengthy, we will structure it in points.

    A11.1 We agree that starting this section with a description of the findings would be an improvement, and we will implement this change in the paper. [Page 9, lines 12–25]

    A11.2 Furthermore, as already agreed, we will minimise the use of abbreviations in the entire paper, to improve clarity and ease of reading.

A11.3 We think that adding the average result for every variable would not be relevant, as there are clearly differences within the four regions. These differences would not be apparent from the average that will be consistently be a low number. However, we will add the average per variable per region, so that the distinction between regions will still be clear. [Page 10, Table 2]

A11.4 Regarding your question on a larger range of variables, we are not sure whether the question relates to the number of variables, or the spanned range of values for the single variables, so we will provide an answer to both interpretations. I) There are some factors that we could not include in the analysis, as stated in the discussion, such as the role of grazers (or other disturbances), and the role of nutrients. However such data are either not available, or their role is still under discussion. Furthermore, to improve our results, new variables must have a strong regional effect, and this effect must not be connected with the one of the variables already considered. For the same reason, the GAM results using all 8 variables, or only the 6 used in latter part of the paper, are very similar, and introducing new variables does not improve the results. Henceforth, we assume that the improvement of additional variables could only be minor. II) The case of larger ranges for the variables would only make sense when considering a larger geographical range. This would at the same time increase the extent of the biome analysed, including areas at mid-latitudes or at more than 70N. Doing so would introduce different plant species and vegetation controls, resulting into a different problem entirely that would, most likely, require a revised set of variables. Thus, it is difficult to make predictions on the outcome of such analysis, but we hypothesise that the increased complexity would not benefit the explained variance.

C12. "Results-vs Discussion: I realize that the above section should not be too much discussion. From page 7, lines 13–15 to page 8, lines 1–20 onwards you have a mix of results and discussion. These should be separated and should have a new section in the discussion chapter."

A12. Dear referee, we understand your point and we will try to implement your suggestion, separating findings and interpretations. [Page 9, lines 26–32; page 10, lines 1–6; page 17, lines 15–33; page 18, lines 1–5]

C13. "Phase-space results: Again take more time to introduce Fig2a. Is it not only the phase-space but also the KDE? Also mention in the text what these intersections are. Mention what the colours are and how we need to interpret Fig2a. After that introduce Fig2b."

A13. Thank you for your valuable suggestion, we will introduce and explain the figures more carefully, improving the captions as well. [Page 10, lines 8–10; page 11, lines 1–2 and 7–17; page 12, Figure 2]

C14. "I also like to see a correlation matrix how the different EVs are correlated with each other. It is now unclear why you show Eastern North Eurasia with the combination of the two and why not a combination of other variables. I think that MAR and mean_TMIN are highly correlated as they are placed on one line, meaning overlap of information."

A14. Following your comment, we will include a correlation matrix in the supplementary material, as we think it would distract the reader from the flow of information. Regarding the figure, it is only meant as an example of the fact that within the regions, for some variables, e.g., precipitation and minimum temperature in Eastern North Eurasia, it is possible to find a clear separation between the three vegetation states (regardless of correlations), whereas for some other pairs, this separation is not clear and we find intersections. Hence, the choice of Northern Eurasia with those specific variables was aimed at exemplifying this point with a figure. We take your point that this information is not clearly conveyed. Hence, we will specify it within the manuscript. [Page 10, lines 8–10; page 11, lines 1–2 and 7–17; page 12, Figure 2]

C15. "3.3: I do not understand the part at page 9, lines 5–10. I can see that you are interested in grid cells having similar EVs but not similar tree cover. However I am confused how to read Table 3, why is that you have four columns? If you mention a number of classes (page 9, line 8), what do you mean?"

A15. We included Table 3 as a summary of the possible vegetation states found during the analysis. However, from your comment we realise that it causes confusion due to its structure. For this reason, we will reshape it, including only three columns and making it clear that they correspond only to the possible monostable, bistable, and fire-disturbed vegetation states. The classes we refer to at page 9, line 8 are the 19 classes reported in Table 4, i.e. the classes that allow for bistable states. To make sure this sentence does not cause confusion, we will rephrase it to explicitly mention it. [Page 11, lines 25–27 and 30–31; page 13, Table 3]

C16. "It is very interesting to see how these data are clustered. I have problems with reading the different colours in the legend. Also some symbols have a black line and others not, but unclear if this relates to the fire or non-fire disturbed states or does it relate to single stable vs bistable data points? Can you also see some spatial patterns of data which have the same bistability, but now currently in a different mode?"

A16. We found hard to retrieve a colourblind-safe colour-scheme with eight colours. However, all the symbols should have the same structure and only different colours. We will try to improve clarity and increase the size of the legend markers. Regarding your second question, with our setup it is only possible to detect states with bistability when they are in a different mode. [Page 13, Figure 3]

C17. "Figure 4: It is Silvermans test (two words)."

A17. You are right. The typo is due to the name of the package used for the implementation, which is the one the plot refers to. We will correct this for consistency and correctness. Thank you for your comment. [Page 15, Figure 4]

C18. "Treeless state: I agree with your statement that tree cover below 20% is difficult to measure wit RS. Therefore I have my doubts about the results of Fig. 4. Why is it that you use in that detail the tree cover fractions below 20%? What do you want to show with these figures. There is not much text about figure 4, so is it needed or can you directly introduce figure 5. Although for this figure, the same holds for the data <20%."

A18. Dear Referee, we understand your confusion about this. We will clarify better in the text and in the captions the meaning of these figures. To answer your questions, the plot in figure 4 and the Silverman's test are meant only to show that there is a clear separation between the modes. And this can be clearly seen when looking at the decrease in frequency happening around 20% tree cover (on both sides). The unsuitability of the MODIS tree cover fraction product below 20% comes into play only when trying to make fine assessments at high resolution. However, in this particular case, only the generic distribution is important, i.e., the fact that there is a peak below 20% tree cover. This information is reliable, as if the tree cover would have been higher, it would have been measured with higher precision by the RS instruments, hence we can conclude that the peak is present. On the other hand, figure 5 is related to the internal variability of the tree cover fraction dataset. The modal peaks are in fact more spread apart than what internal variability alone could cause, making them significant. Again, the precise distribution below 20% tree cover is not extremely important, only the fact that it is below such threshold. [Page 12, lines 10–14; page 15, lines 1–4, 6–7, and 9–11; page 15, Figure 4; page 16, Figure 5]

C19. "Do not understand your statement on page 14, line 6; what kind of feedbacks? You didnt study this, so why is it that they might or might not play a role?"

A19. We are referring to the main feedbacks happening between vegetation, environmental variables, and climate. However, we understand your point that we did not discuss them. For this reason, we will include details of the main feedbacks within the introduction and within the expanded description of the environmental variables. So that we can make a clearer reference to them. [Page 2, lines 19–29; page 4, lines 7–34; page 5, lines 1–14]

C20. "I found the discussion on N-cycling, decomposition, fertilisation a bit too much detail in comparison to the work you have presented. You have now linked it to soil type, and if you would be more interested in Nitrogen then you could have used modelled maps from DGVMs (as LPJ-Guess) or use maps or soil organic matter. I think that there are more important things to discuss, for instance why individual set of EVs are different between the regions, why fire is that important, which regions are more sensitive to a change in temperature then others, or a change in permafrost depth etc. etc. So keep the discussion more related to your own findings."

A20. Dear Referee, this part of the discussion was intended to show that there are other factors playing a role in the boreal forest biome. However, we will implement your suggestion by reducing its extent and by expanding the discussion related to our findings. Additionally, we will move here part of the discussion presented in the GAMs results section. [Page 17, lines 15–33; page 18, lines 1–5, 14–16, and 27–35; page 19, lines 1–5]

**Reply to Anonymous Referee #3**

B. Abis and V. Brovkin

Dear Referee,

Thank you for your positively constructive review and for your insightful comments. In this document, we will provide an answer to your comments and queries.

Best regards,
B. Abis and V. Brovkin

C1. "Please add citations to the sentence '. . . and to ecosystems exhibiting potential alternative tree cover states under the same environmental conditions. . . ' on page 2, line 5."

A1. We will add citations. [Page 2, line 8]

C2. "Table 2, 4 contains many acronyms without complete name in the table caption. The figures or tables should stand alone. Suggest to add complete names to table captions or to the table. What do you mean by 'very low, medium low, very high and medium high' in Table 4? Please quantify or specify them."

A2. Dear Referee, to make Table 4 more readable, we initially thought to include only qualitative information, introduced in the text, and to include the actual ranges only in the supplementary material. However, we will change this and make Table 4 more complete, with all the ranges and the number of gridcells per class. Furthermore, as you suggested, we will expand the captions to make the tables and the figures stand alone. [Page 6, Table 1; page 8, Figure 1; page 10, Table 2; page 12, Figure 2; page 13, Table 3; page 14, Table 4; page 14, Figure 3; page 15, Figure 4; page 16, Figure 5]

C3. "The results or findings in abstract are too general, e.g. "we find that environmental conditions exert a strong control. . . ", "we find that the relationship between tree cover and environment is different within the four boreal regions. . . ", and etc. Please be specific."

A3. We will reformulate the abstract to make it more readable, with clear and specific statements about our findings. [Page 1, lines 5, 9–10, and 14–15]

C4. "Conclusions (lines 2–29 on page 15) are too long, more like summary. Please shorten this section."

A4. Dear Referee, as you suggested, we will shorten the conclusions by limiting the discussion and generalisations in them and keeping only our key findings. [Page 19, lines 23–32; page 20, lines 1–10]

[revised manuscript text omitted]
 mean minimum temperature and MAR mean annual rainfall (a), and MSSM mean spring soil moisture and GDD0 growing degree days above $0°C$ (b), for Eastern North Eurasia. Vegetation states are colour-coded as follows: green for forest, orange for open woodland, and purple for treeless. The isolines describe the probability of finding the three vegetation states under the specified EVs environmental variables regimes, with intense colours indicating higher probabilities. Intersections Highlighted intersections in phase-space represent areas with different vegetation states under the same environmental conditions (a), whereas the marked areas with only one dominant state hint at the unimodality of the underlying distribution (b). Marginal distributions for the variables are reported to the sides of the plots in the form of histograms.

Eurasia, and consist of 5 EVs environmental variables classes, of which 4 are also related to equivalent tree cover states. All 19 classes are reported in Table 4. Qualitative indexes for the EVs environmental variables, except for ST and PZI, represent the bin into which the variable's value falls in the classification, as described in Section 2.2; the order is: very low, low, medium-low, medium-high, high, very high. Precise values are reported in Table 4 (see Supplementary Material for further details). Soil texture is described as belonging to the sand, loam, or clay group. Permafrost is described as sparse, discontinuous, frequent, or continuous. Each EV environmental variable class contains two possible vegetation states, e.g., forest and open woodland, that are consistently found under the same specified environmental regimes.

Table 4 and Figure 3 pinpoint the conditions and locations, respectively, of the possible alternative tree cover states in the boreal area. Results of the To test whether the distributions of the possible alternative tree cover states are multimodal, we employ the Silverman's tests on the test. Each Silverman's test assesses the hypothesis that the number of modes of the distributions of the alternative open woodland and treeless gridcells, and the one of the alternative open woodland and forest gridcells confirm their bimodality, as shown in Fig. 4. Each Silverman's test assesses the hypothesis that the number of modes of each distribution , is $\leq 1$. The tests show that the minimum number of modes to describe the distributions is two, for both cases, is two, with $p$-values smaller than 0.001 and 0.01 0.001 and 0.01, respectively. Furthermore, by fitting these distributions Figure 4 shows the results of the Silverman's tests on the distributions of the possible alternative tree cover states, confirming their bimodality, together with the respective tree

**Table 3.** Summary of possible vegetation states, divided as monostable, bistable, and fire disturbed. Fire disturbed states have a higher fire regime than the indicated counterpart. Type of state Single stable Treeless (TCFTreeless always refers to TCF<20%) Open %, open woodland (to 20%≤TCF<45%) Forest (TCF%, and forest to TCF≥45%) %.

[revised manuscript text omitted]

---

## Author Response (AR2)

**Reply to the Associate Editor**

**B. Abis and V. Brovkin**

Dear Editor,

Thank you for all the attentions dedicated to our manuscript submission. In this document, we will answer your last comments regarding the supplementary material, providing the updated version of the file Supplementary_Tables.pdf.

Best regards,
B. Abis and V. Brovkin

Q1. "Supplementary_Tables.pdf. For clarification, please use figure and table numbers different from those in the main text: for example, Figure S1 and Table S1."

A1. We implemented this correction.

Q2. "Supplementary_Tables.pdf. Please explain the colors (green and purple) used in Figure 1 (looks like a table) in the caption."

A2. We realised this table was somewhat difficult to interpret, so we decided to split it into two summary tables, with more detailed captions. Furthermore, the table was named as a figure due to its formatting. By splitting the table, we were able to simplify its formatting and include everything as tables. The colours were indicative of regions and overlaps in case of fire disturbed states, in which one of the two states might be present in another class not fire disturbed. This information was used to create the corrected totals for each case. We decided to remove the colours to improve readability, and we included the necessary information in the captions.

**Supplementary Tables to "Environmental Conditions for Alternative Tree Cover States in High Latitudes"**

Beniamino Abis[1,2] and Victor Brovkin[2]

[1]International Max Planck Research School on Earth System Modelling, Hamburg, Germany
[2]Max Planck Institute for Meteorology, Hamburg, Germany

**Table S1.** Correlation matrix among all the environmental variables across North America.

| Correlation matrix for Eastern North America | | | | | | | | |
|---|---|---|---|---|---|---|---|---|
| | MAR | MSSM | MTmin | PZI | FF | GDD0 | PTD | ST |
| MAR | 1.0000 | 0.8891 | 0.8253 | -0.7847 | 0.1039 | 0.6490 | 0.3097 | -0.0213 |
| MSSM | 0.8891 | 1.0000 | 0.7148 | -0.6763 | 0.0503 | 0.5574 | 0.2356 | -0.0173 |
| MTmin | 0.8253 | 0.7148 | 1.0000 | -0.9295 | 0.2115 | 0.9269 | 0.5796 | 0.0303 |
| PZI | -0.7847 | -0.6763 | -0.9295 | 1.0000 | -0.2830 | -0.9032 | -0.5726 | -0.0009 |
| FF | 0.1039 | 0.0503 | 0.2115 | -0.2830 | 1.0000 | 0.2539 | 0.3054 | 0.0610 |
| GDD0 | 0.6490 | 0.5574 | 0.9269 | -0.9032 | 0.2539 | 1.0000 | 0.6239 | -0.0190 |
| PTD | 0.3097 | 0.2356 | 0.5796 | -0.5726 | 0.3054 | 0.6239 | 1.0000 | 0.0787 |
| ST | -0.0213 | -0.0173 | 0.0303 | -0.0009 | 0.0610 | -0.0190 | 0.0787 | 1.0000 |

| Correlation matrix for Western North America | | | | | | | | |
|---|---|---|---|---|---|---|---|---|
| | MAR | MSSM | MTmin | PZI | FF | GDD0 | PTD | ST |
| MAR | 1.0000 | 0.7899 | 0.5708 | -0.5321 | -0.0081 | 0.3072 | 0.2775 | -0.1028 |
| MSSM | 0.7899 | 1.0000 | 0.4975 | -0.4778 | 0.0286 | 0.2183 | 0.2208 | -0.0638 |
| MTmin | 0.5708 | 0.4975 | 1.0000 | -0.8895 | 0.2362 | 0.8605 | 0.7730 | -0.1384 |
| PZI | -0.5321 | -0.4778 | -0.8895 | 1.0000 | -0.2610 | -0.6850 | -0.6206 | 0.0677 |
| FF | -0.0081 | 0.0286 | 0.2362 | -0.2610 | 1.0000 | 0.2557 | 0.2153 | -0.0829 |
| GDD0 | 0.3072 | 0.2183 | 0.8605 | -0.6850 | 0.2557 | 1.0000 | 0.8225 | -0.1951 |
| PTD | 0.2775 | 0.2208 | 0.7730 | -0.6206 | 0.2153 | 0.8225 | 1.0000 | -0.2866 |
| ST | -0.1028 | -0.0638 | -0.1384 | 0.0677 | -0.0829 | -0.1951 | -0.2866 | 1.0000 |

**Table S2.** Correlation matrix among all the environmental variables across North Eurasia.

| | MAR | MSSM | MTmin | PZI | FF | GDD0 | PTD | ST |
|---|---|---|---|---|---|---|---|---|
| | | | Correlation matrix for Eastern North Eurasia | | | | | |
| MAR | 1.0000 | 0.8289 | 0.5571 | -0.5170 | 0.0243 | 0.3526 | 0.0627 | -0.0238 |
| MSSM | 0.8289 | 1.0000 | 0.5134 | -0.4385 | -0.1163 | 0.2293 | 0.0242 | -0.1577 |
| MTmin | 0.5571 | 0.5134 | 1.0000 | -0.8917 | 0.3394 | 0.7816 | 0.3576 | -0.3114 |
| PZI | -0.5170 | -0.4385 | -0.8917 | 1.0000 | -0.3712 | -0.7641 | -0.3359 | 0.3286 |
| FF | 0.0243 | -0.1163 | 0.3394 | -0.3712 | 1.0000 | 0.5002 | 0.3976 | -0.1153 |
| GDD0 | 0.3526 | 0.2293 | 0.7816 | -0.7641 | 0.5002 | 1.0000 | 0.5128 | -0.3454 |
| PTD | 0.0627 | 0.0242 | 0.3576 | -0.3359 | 0.3976 | 0.5128 | 1.0000 | -0.3295 |
| ST | -0.0238 | -0.1577 | -0.3114 | 0.3286 | -0.1153 | -0.3454 | -0.3295 | 1.0000 |
| | | | Correlation matrix for Western North Eurasia | | | | | |
| | MAR | MSSM | MTmin | PZI | FF | GDD0 | PTD | ST |
| MAR | 1.0000 | 0.8076 | 0.0038 | -0.2666 | -0.6025 | -0.2109 | -0.2476 | -0.0190 |
| MSSM | 0.8076 | 1.0000 | -0.0999 | -0.0737 | -0.5793 | -0.2450 | -0.2464 | -0.0636 |
| MTmin | 0.0038 | -0.0999 | 1.0000 | -0.7889 | 0.5027 | 0.9316 | 0.7961 | -0.0954 |
| PZI | -0.2666 | -0.0737 | -0.7889 | 1.0000 | -0.2001 | -0.6642 | -0.4717 | 0.1565 |
| FF | -0.6025 | -0.5793 | 0.5027 | -0.2001 | 1.0000 | 0.6525 | 0.6215 | -0.0617 |
| GDD0 | -0.2109 | -0.2450 | 0.9316 | -0.6642 | 0.6525 | 1.0000 | 0.8325 | -0.0281 |
| PTD | -0.2476 | -0.2464 | 0.7961 | -0.4717 | 0.6215 | 0.8325 | 1.0000 | -0.0831 |
| ST | -0.0190 | -0.0636 | -0.0954 | 0.1565 | -0.0617 | -0.0281 | -0.0831 | 1.0000 |

**Table S3.** Boundaries of the bins used in the classification.

| | Eastern North America | | | | | | |
| --- | --- | --- | --- | --- | --- | --- | --- |
| | 0 | 1 | 2 | 3 | 4 | 5 | 6 |
| Mtmin 1x | -16.9697 | -6.5599 | -4.5902 | -2.6206 | -0.6509 | 1.3187 | 5.8752 |
| MSSM 6x | 104.2174 | 238.6162 | 301.3687 | 364.1211 | 426.8735 | 489.6259 | 598.2341 |
| Mar 36x | 120.0692 | 534.9277 | 647.2662 | 759.6046 | 871.9431 | 984.2815 | 1,607.1846 |
| PZI 216x | 0.0000 | 0.0878 | 0.1757 | 0.6411 | 1.0000 | | |
| ST 1000x | 1,2,12 | 3,4,5,6,7,8 | 9,10,11,13,14 | | | | |
| FF 3864x | 0.0000 | 0.0714 | 0.1429 | 0.3714 | 1.8824 | | |

| | Western North America | | | | | | |
| --- | --- | --- | --- | --- | --- | --- | --- |
| | 0 | 1 | 2 | 3 | 4 | 5 | 6 |
| Mtmin 1x | -15.2733 | -8.9262 | -6.2897 | -3.6531 | -1.0165 | 1.6200 | 8.4559 |
| MSSM 6x | 19.2895 | 137.0854 | 188.1259 | 239.1663 | 290.2067 | 341.2471 | 694.4776 |
| Mar 36x | 51.8308 | 191.7554 | 284.9050 | 378.0546 | 471.2042 | 564.3538 | 3138.8692 |
| PZI 216x | 0.0000 | 0.2012 | 0.4024 | 0.8197 | 1.0000 | | |
| ST 1000x | 1,2,12 | 3,4,5,6,7,8 | 9,10,11,13,14 | | | | |
| FF 3864x | 0.0000 | 0.2941 | 0.5882 | 0.7059 | 2.9412 | | |

| | Eastern North Eurasia | | | | | | |
| --- | --- | --- | --- | --- | --- | --- | --- |
| | 0 | 1 | 2 | 3 | 4 | 5 | 6 |
| Mtmin 1x | -17.9223 | -10.5526 | -8.5407 | -6.5288 | -4.5170 | -2.5051 | 3.0637 |
| MSSM 6x | 54.3041 | 155.1053 | 199.3099 | 243.5146 | 287.7192 | 331.9239 | 573.2997 |
| Mar 36x | 132.3769 | 331.0546 | 399.7775 | 468.5004 | 537.2233 | 605.9462 | 1006.1846 |
| PZI 216x | 0.0000 | 0.0050 | 0.0100 | 0.9662 | 1.0000 | | |
| ST 1000x | 1,2,12 | 3,4,5,6,7,8 | 9,10,11,13,14 | | | | |
| FF 3864x | 0.0000 | 0.4118 | 0.8235 | 0.9412 | 3.5294 | | |

| | Western North Eurasia | | | | | |
| --- | --- | --- | --- | --- | --- | --- |
| | 0 | 1 | 2 | 3 | 4 | 5 |
| Mtmin 1x | -8.2882 | -4.7622 | -2.4858 | -0.2094 | 2.0669 | 5.3837 |
| MSSM 5x | 99.4328 | 255.6825 | 291.2951 | 326.9077 | 362.5203 | 440.3554 |
| Mar 25x | 204.7615 | 520.0138 | 567.5974 | 615.1810 | 662.7646 | 797.2000 |
| PZI 125x | 0.0000 | 0.0705 | 0.1410 | 0.2774 | | |
| ST 1000x | 1,2,12 | 3,4,5,6,7,8 | 9,10,11,13,14 | | | |
| FF 3864x | 0.0000 | 0.2647 | 0.5294 | 0.5882 | 3.1765 | |

Table S4. Summary of possible alternative classes. The multipliers refer to the boundaries in Table S3. The vegetation states correspond to the sum of the multipliers. In the fire disturbed cases, the total number of gridcells is corrected to take into account overlaps with cases where the same vegetation state is also present among the states which are not fire disturbed.

| Vegetation State | | 3864x | 1000x | 216x | 36x | 6x | 1x | # Gridcells I | # Gridcells II | Total |
|---|---|---|---|---|---|---|---|---|---|---|
| Eastern North America | | | | | | | | | | |
| Treeless – Open woodland | 496 | 0 | 0 | 2 | 1 | 4 | 4 | 12 | 12 | **24** |
| Treeless – Open woodland | 669 | 0 | 0 | 3 | 0 | 3 | 3 | 13 | 7 | **20** |
| Forest – Open woodland | 216 | 0 | 0 | 0 | 5 | 5 | 6 | 40 | 18 | **58** |
| Western North America | | | | | | | | | | |
| Forest – Open woodland | 4988 | 1 | 1 | 0 | 3 | 2 | 4 | 18 | 9 | **27** |
| Forest – Open woodland | 6988 | 1 | 3 | 0 | 3 | 2 | 4 | 29 | 15 | **44** |
| Eastern North Eurasia | | | | | | | | | | |
| Treeless – Open woodland | 1539 | 0 | 1 | 2 | 2 | 5 | 5 | 24 | 11 | **35** |
| Treeless – Open woodland | 5519 | 1 | 1 | 3 | 0 | 1 | 1 | 27 | 7 | **34** |
| Forest – Open woodland | 1519 | 0 | 1 | 2 | 2 | 2 | 3 | 13 | 10 | **23** |
| Forest – Open woodland | 1575 | 0 | 1 | 2 | 3 | 5 | 5 | 12 | 11 | **23** |
| Forest – Open woodland | 5439 | 1 | 1 | 2 | 3 | 5 | 5 | 11 | 10 | **21** |
| Forest – Open woodland | 5469 | 1 | 1 | 2 | 4 | 4 | 5 | 11 | 8 | **19** |
| Open woodland – FD Treeless | 1655 | 0 | 1 | 3 | 0 | 1 | 1 | 10 | 58 | **68** |
| Open woodland – FD Treeless | 5519 | 1 | 1 | 3 | 0 | 1 | 1 | 7 | 35 | **35** |
| Open woodland – FD Forest | 1575 | 0 | 1 | 2 | 3 | 5 | 5 | 11 | 11 | **11** |
| Forest – FD Open woodland | 1519 | 0 | 1 | 2 | 2 | 2 | 3 | 13 | 11 | **11** |
| Forest – FD Open woodland | 1575 | 0 | 1 | 2 | 3 | 5 | 5 | 12 | 17 | **17** |

| Vegetation State | | 3864x | 1000x | 125x | 25x | 5x | 1x | # Gridcells I | # Gridcells II | Total |
|---|---|---|---|---|---|---|---|---|---|---|
| Western North Eurasia | | | | | | | | | | |
| Treeless – Open woodland | 16532 | 4 | 1 | 0 | 3 | 0 | 1 | 33 | 7 | **40** |
| Forest – Open woodland | 69 | 0 | 0 | 0 | 2 | 3 | 4 | 11 | 7 | **18** |
| Forest – Open woodland | 1093 | 0 | 1 | 0 | 3 | 3 | 3 | 12 | 8 | **20** |

Table S5. Total amount of gridcells related to alternative classes.

| NA_E Total | NA_W Total | NA Total | NA % | EA_E Total | EA_W Total | EA Total | EA % | Global Total | Global % |
|---|---|---|---|---|---|---|---|---|---|
| 102 | 71 | 173 | 3.55 | 297 | 78 | 375 | 5.06 | 548 | 4.46 |